# An in vitro stem cell model of human epiblast and yolk sac interaction

**Kirsty ML Mackinlay[1], Bailey AT Weatherbee[1], Viviane Souza Rosa[1,2,3], Charlotte E Handford[1,4], George Hudson[1], Tim Coorens[5], Lygia V Pereira[2], Sam Behjati[5], Ludovic Vallier[6], Marta N Shahbazi[1,3]\*, Magdalena Zernicka-Goetz[1,7]\***

[1]Mammalian Embryo and Stem Cell Group, University of Cambridge, Department of Physiology, Development and Neuroscience, Cambridge, United Kingdom; [2]National Laboratory for Embryonic Stem Cells (LaNCE), Department of Genetics and Evolutionary Biology, Institute of Biosciences, University of São Paulo, São Paulo, Brazil; [3]MRC Laboratory of Molecular Biology, Cambridge Biomedical Campus, Cambridge, United Kingdom; [4]Centre for Trophoblast Research, University of Cambridge, Cambridge, United Kingdom; [5]Wellcome Sanger Institute, Cambridge, United Kingdom; [6]Wellcome – MRC Cambridge Stem Cell Institute, Cambridge Biomedical Campus, Cambridge, United Kingdom; [7]Synthetic Mouse and Human Embryology Group, California Institute of Technology (Caltech), Division of Biology and Biological Engineering, Pasadena, United States

**\*For correspondence:**
mshahbazi@mrc-lmb.cam.ac.uk (MNS);
mz205@cam.ac.uk (MZ-G)

**Competing interests:** The authors declare that no competing interests exist.

**Abstract** Human embryogenesis entails complex signalling interactions between embryonic and extra-embryonic cells. However, how extra-embryonic cells direct morphogenesis within the human embryo remains largely unknown due to a lack of relevant stem cell models. Here, we have established conditions to differentiate human pluripotent stem cells (hPSCs) into yolk sac-like cells (YSLCs) that resemble the post-implantation human hypoblast molecularly and functionally. YSLCs induce the expression of pluripotency and anterior ectoderm markers in human embryonic stem cells (hESCs) at the expense of mesoderm and endoderm markers. This activity is mediated by the release of BMP and WNT signalling pathway inhibitors, and, therefore, resembles the functioning of the anterior visceral endoderm signalling centre of the mouse embryo, which establishes the anterior-posterior axis. Our results implicate the yolk sac in epiblast cell fate specification in the human embryo and propose YSLCs as a tool for studying post-implantation human embryo development *in vitro*.

## Introduction

Over the first 5 days of human development, the fertilised zygote undergoes a series of cleavage divisions and morphogenetic events that give rise to a blastocyst, which comprises an inner cell mass (ICM) and a fluid-filled cavity surrounded by the trophoblast, the extra-embryonic tissue that forms the placenta (*Edwards et al., 1970*; *Steptoe et al., 1971*). As the blastocyst matures, the ICM undergoes a morphological sorting process which leads to the specification of the epiblast, progenitor of the embryo proper, and the hypoblast (extra-embryonic endoderm), the precursor tissue of the yolk sac (*Wamaitha and Niakan, 2018*).

Once the hypoblast, epiblast, and trophoblast have been specified, the human blastocyst is ready to implant into the uterine wall of the mother on embryonic day 6–7 (E6–7) post-fertilisation (*Steptoe et al., 1971*). *In vivo* and *in vitro* studies of human and non-human primates (*Deglincerti et al., 2016*; *Luckett, 1975*; *Ma et al., 2019*; *Niu et al., 2019*; *Shahbazi et al.,*

*2016*) have shown that upon implantation, epiblast cells polarise and undergo lumenogenesis to form the amniotic cavity. Epiblast cells located in close proximity to the hypoblast become a columnar epiblast disc, in contact with the amniotic cavity. The hypoblast proliferates, giving rise to the primary yolk sac on day 10. Around day 14, the primitive streak forms posteriorly in the epiblast, triggering the onset of gastrulation (*Sasaki et al., 2016*; *Shahbazi, 2020*).

The signalling pathways that govern early post-implantation morphogenesis in the human embryo are not fully understood. In the mouse embryo, a group of asymmetrically localised cells, collectively known as the anterior visceral endoderm (AVE), is established at E5.5, which secrete NODAL, WNT, and BMP antagonists (e.g. Lefty, Dkk1, Cer1, Noggin), leading to the formation of a gradient of signalling activity across the anterior-posterior axis of the embryo (*Beddington and Robertson, 1999*; *Rivera-Pérez and Magnuson, 2005*). Interestingly, AVE-like cells have also been identified in the cynomolgus monkey yolk sac *in vivo* (*Sasaki et al., 2016*), and in human embryos cultured *in vitro* (*Molè et al., 2021*; *Xiang et al., 2020*). However, the functional relevance of these descriptive findings is not yet known.

Culture conditions now support the development of human embryos *in vitro* up until the internationally accepted limit of 14 days (*Deglincerti et al., 2016*; *Shahbazi et al., 2016*; *Xiang et al., 2020*; *Zhou et al., 2019*). Nonetheless, the reliance on surplus *in vitro* fertilised (IVF) embryos, combined with the ethical considerations associated with the genetic manipulation of human embryos, means that there are currently barriers to the study of post-implantation human development. Accordingly, there is a need to create stem cell models of post-implantation embryogenesis, thereby producing a system that is amenable to genetic and molecular manipulation. Such research has already begun, with human embryonic stem cells (hESCs) being used to model early embryogenesis (*Etoc et al., 2016*; *Moris et al., 2020*; *Shahbazi et al., 2017*; *Simunovic et al., 2019*; *Warmflash et al., 2014*). However, the current systems lack the inclusion of extra-embryonic tissues, which are likely key regulators of human epiblast patterning. Thus, to determine how intercellular communication drives post-implantation human embryogenesis, it is important to establish an *in vitro* model that encompasses both embryonic and extra-embryonic lineages.

Here, we aimed to build an *in vitro* model that would allow us to study the interactions between the yolk sac and epiblast of the post-implantation human embryo. To do this, we first generated yolk sac-like cells (YSLCs) by simultaneously stimulating ACTIVIN-A, WNT, and JAK/STAT signalling in peri-implantation-like human pluripotent stem cells (hPSCs) (*Gafni et al., 2013*). We then modelled the interaction between these YSLCs and hESCs, which revealed that YSLCs induce the expression of pluripotency and anterior ectoderm markers at the expense of mesoderm and endoderm markers in hESCs by inhibiting BMP and WNT signalling pathways. Our results indicate that these YSLCs can be used as a tool for studying how the interaction between human embryonic and extra-embryonic cells regulates human embryo development.

## Results

### ACTIVIN-A, WNT, and JAK/STAT pathway activation induces an endodermal fate in hPSCs

To model epiblast-yolk sac crosstalk, we sought *in vitro* counterparts of each of these tissues. Conventionally, hESCs have been used as a reliable model of the human epiblast, harbouring a post-implantation, 'primed' pluripotent state (*Nakamura et al., 2016*; *Shao et al., 2017a*; *Simunovic et al., 2019*; *Warmflash et al., 2014*; *Zheng et al., 2019*). To derive a YSLC, we intended to use a starting cell line with extra-embryonic competency. We hypothesised that human extended pluripotent stem cells (hEPSCs) were an appropriate candidate as they were shown to have extra-embryonic potential when incorporated into mouse embryos (*Yang et al., 2017*).

We first conducted a screen by adding pairs of signalling pathway activators that have been implicated in murine primitive endoderm (hypoblast) development to hEPSC medium (LCDM medium, see Materials and methods) (*Figure 1—figure supplement 1A*). hEPSCs were derived from the RUES2-GLR reporter line which reports for expression of SOX2 (pluripotent epiblast), SOX17 (endoderm), and BRACHYURY (mesoderm) (*Martyn et al., 2018*). By analysing SOX17 and SOX2 reporter fluorescence intensity, we discerned that the combination of WNT (via the GSK3 activator CHIR 99021) and ACTIVIN-A pathway activation led to the greatest upregulation of SOX17 at the expense

of SOX2 after 6 days (*Figure 1—figure supplement 1B,C*). This result is in line with ACTIVIN-A and WNT signalling being involved in the specification of mouse, monkey, and human primitive endoderm *in vitro* and *in vivo* (*Anderson et al., 2017*; *Blakeley et al., 2015*; *Boroviak et al., 2015*; *Linneberg-Agerholm et al., 2019*). We further optimised the medium by increasing the concentrations of both CHIR 99021 and ACTIVIN-A (*Figure 1—figure supplement 1D,E*), and removing the other cytokines from the LCDM medium apart from human leukemia inhibitory factor (LIF), which we found to be necessary for proliferation during the conversion (*Figure 1—figure supplement 1F*). Henceforth, the medium consisting of N2B27 supplemented with 100 ng ml$^{-1}$ human ACTIVIN-A, 3 µM CHIR99021, and 10 ng ml$^{-1}$ recombinant human LIF is referred to as 'ACL'.

To further assess the effect of ACL treatment on hEPSCs after 6 days of culture, we analysed the expression levels of SOX17 and SOX2 using reporter fluorescence at the single cell level. This revealed a strong shift from an originally SOX2 high, SOX17 low population of cells, to the converse (*Figure 1A–C*). Furthermore, gene expression analysis showed that pan-endodermal markers *SOX17, GATA6*, and *EOMES* were all upregulated following ACL treatment (*Figure 1D*, and *Figure 1—figure supplement 2A*). Despite observed downregulation of *SOX2* expression, the expression of *OCT4* and *NANOG* increased (*Figure 1D* and *Figure 1—figure supplement 2A*), which has previously been observed in hESCs overexpressing SOX17 (*Séguin et al., 2008*). The inability of ACL to fully downregulate pluripotency markers in hPSCs may indicate that cells may still preserve some features of pluripotent cells.

To determine whether ACL could also induce endodermal fate in other hPSCs, we treated both RUES2-GLR Rset hPSCs, which harbour a pluripotent state that is intermediate between pre-implantation 'ground-state' naïve pluripotency and post-implantation lineage-biased primed pluripotency (*Gafni et al., 2013*; *Kilens et al., 2018*), and conventional primed RUES2-GLR hESCs, that are similar to the post-implantation human epiblast (*Molè et al., 2021*; *Nakamura et al., 2016*). After 6 days of treatment, endodermal markers were upregulated and SOX2 was downregulated in both Rset hPSCs and hESCs (*Figure 1E–K*). This expression pattern was also observed after 1 month of culture (*Figure 1—figure supplement 3*). We also observed the requirement of LIF for continued proliferation for both of these treated hPSC populations (*Figure 1—figure supplement 1F*). To evaluate whether these results were cell line specific, we repeated ACL treatment using H9-derived and SHEF6-derived hEPSCs, Rset hPSCs, and hESCs. This revealed a similar result, in that SOX17, GATA6, and EOMES were upregulated after 6 days in all three populations for all cell lines at the protein and RNA level (*Figure 1—figure supplement 2B–K* and *Figure 1—figure supplement 4*). SOX2 protein levels were also downregulated upon ACL treatment in all three hPSCs for all cell lines (*Figure 1—figure supplement 2B–J* and *Figure 1—figure supplement 4*). However, *SOX2* was upregulated at the mRNA level in treated H9 hEPSC cells (*Figure 1—figure supplement 2K*). Single-cell fluorescence intensity analysis revealed this was likely due to a subpopulation of cells that were GATA6 low but expressed high SOX2 levels (*Figure 1—figure supplement 2D*), rather than GATA6 high cells co-expressing high levels of SOX2. This suggests that there is a heterogeneous response to ACL treatment within H9 hEPSCs.

Given that all three hPSC ACL-treated populations upregulated expression of selected endodermal markers, we sought to determine whether ACL treatment led to global gene expression shifts from an epiblast to an endodermal fate – either towards the human hypoblast or the embryonic endoderm lineage, the definitive endoderm. Accordingly, we compared the transcriptome of sorted SOX17::H2B-tdTomato+ converted populations (*Figure 1—figure supplement 5*), along with Rset hPSCs, primed hESCs (*Linneberg-Agerholm et al., 2019*), and hEPSCs (*Yang et al., 2017*), to the lineages of the early human embryo and to human definitive endoderm. To do this, we first identified genes that were differentially expressed between epiblast, trophoblast, and hypoblast throughout pre- or early post-implantation development (E6–14), and *in vitro*-derived definitive endoderm (*Supplementary file 1*; *Cuomo et al., 2020*; *Molè et al., 2021*; *Xiang et al., 2020*; *Zhou et al., 2019*). Markers were identified in a pairwise manner, where lineages were individually compared, and gene signatures for each lineage were defined as those positive markers which were common across all pairwise comparisons. These gene signatures were then incorporated into a gene set-enrichment-based gene set variation analysis (GSVA) that is particularly robust in extrapolating cell type labels from single-cell datasets to bulk RNA-seq datasets (*Diaz-Mejia et al., 2019*; *Hänzelmann et al., 2013*). Coefficients were calculated across each lineage and normalized to 1. Importantly, given that different gene sets are used for each lineage within this analysis,

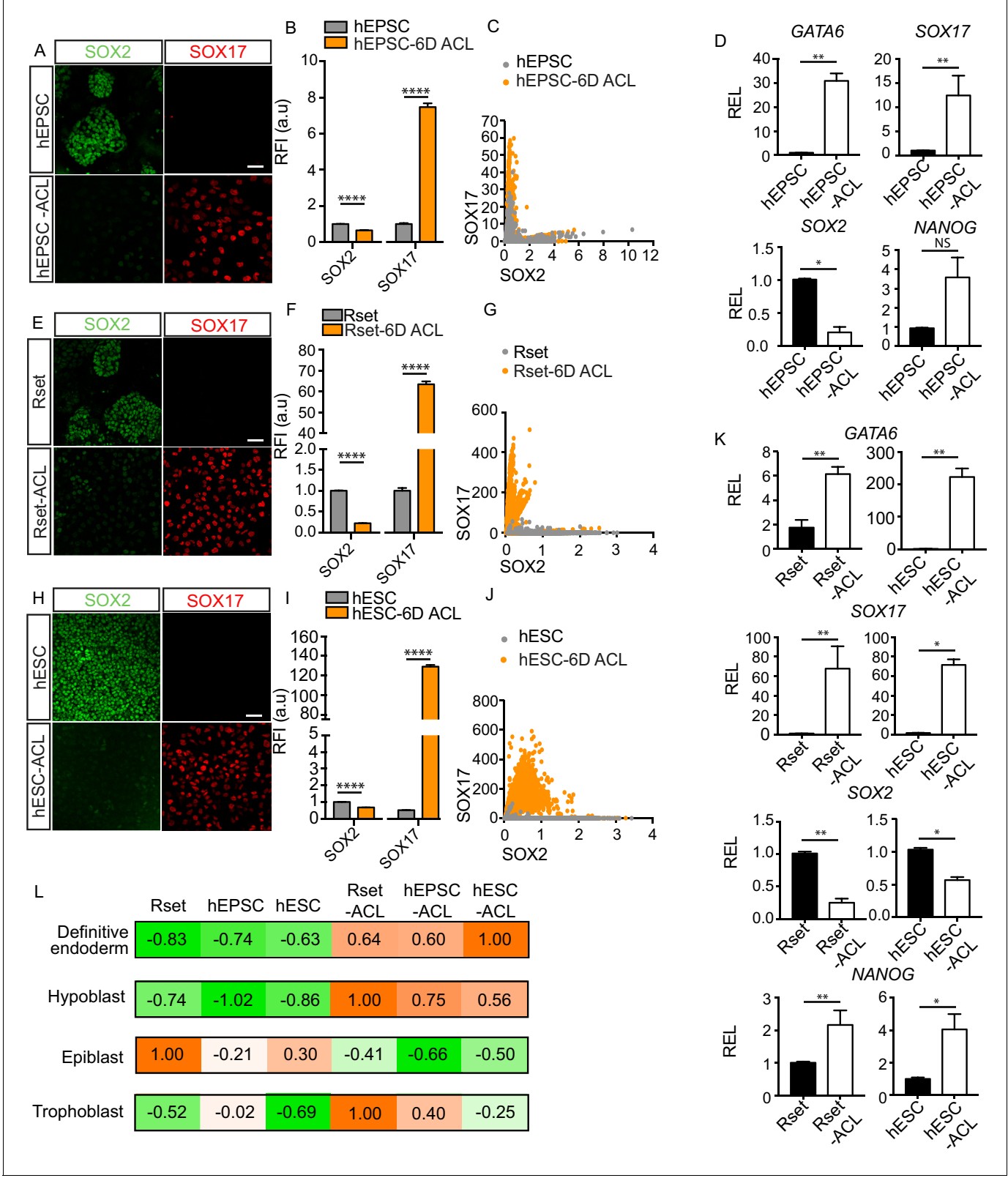

**Figure 1.** Endoderm conversion of RUES2-GLR human pluripotent stem cells (hPSCs) in response to ACL treatment. (**A, E, and H**) Immunofluorescence images of SOX2 (green) and SOX17 (red) in human extended pluripotent stem cells (hEPSCs), Rset human pluripotent stem cells (hPSCs), and human embryonic stem cells (hESCs) after 6 days (6D) of ACL treatment (scale bar = 50 μm). (**B, F, and I**) Bar chart of elative fluorescence intensity (RFI arbitrary units [a.u.]) of SOX17 and SOX2 (± SEM) before and after 6D of ACL treatment in hPSCs as measured via reporter protein immunostaining. Mann–

*Figure 1 continued on next page*

*Figure 1 continued*

Whitney U-test; ****p < 0.0001 (number of cells analysed: hEPSC control n=2736, hEPSC + 6D ACL n=2270, Rset control n=2749, Rset + 6D ACL n=3146, hESC control n=1939, hESC + 6D ACL n=2538, three independent experiments each with one sample). (C, G, and J) Quantification of SOX2 and SOX17 protein levels at the single cell level in hEPSC, Rset hPSC, and hESC 6D ACL-treated cells based on relative reporter fluorescence intensity. Each dot represents an individual cell. (D) Relative expression levels (REL) (± SEM) of *GATA6, SOX17, NANOG* (n=6 samples, three independent experiments) *and SOX2* (n=4 samples, two independent experiments) in hEPSC before and after 6D ACL treatment, normalised to their respective control. Mann–Whitney U-test *p < 0.05, **p < 0.01, ***p < 0.001, ****p < 0.0001. (K) Relative expression levels (REL) (± SEM) of *GATA6, SOX17, SOX2,* and *NANOG* in Rset hPSCs (n=6 samples, three independent experiments), or hESCs (n=4 samples, two independent experiments) after 6D ACL treatment, normalised to their respective control . Mann–Whitney U-test *p < 0.05, **p < 0.01, ***p < 0.001, ****p < 0.0001. (L) Transcriptomic-signature score (gene set variation analysis ([GSVA]) positive marker score) when comparing ACL-treated cells to either the human hypoblast, trophoblast, or epiblast within the human embryo (E6–14), or to human definitive endoderm using sc-RNA-seq expression data (embryo and definitive endoderm) and bulk RNA-seq expression data (cell lines). The higher the value (orange), the more relatively similar to each lineage.

The online version of this article includes the following source data and figure supplement(s) for figure 1:

**Source data 1.** Fluorescence intensity and qPCR analysis of endoderm and pluripotency markers in RUES2 human pluripotent stem cells (hPSCs) in response to ACL.

**Figure supplement 1.** 6-Day molecular screen using RUES2 human pluripotent stem cells (hPSCs).

**Figure supplement 1—source data 1.** Fluorescence intensity analysis of human extended pluripotent stem cell (hEPSC) screen.

**Figure supplement 2.** Endoderm conversion of RUES2 and H9 human pluripotent stem cells (hPSCs) in response to ACL treatment.

**Figure supplement 2—source data 1.** Fluorescence intensity and qPCR analysis of endoderm and pluripotency markers in RUES2 and H9 human pluripotent stem cells (hPSCs) in response to ACL.

**Figure supplement 3.** Prolonged culture of Rset-ACL.

**Figure supplement 3—source data 1.** Fluorescence intensity and qPCR analysis of long-term culture of Rset-ACL.

**Figure supplement 4.** Endoderm conversion of SHEF6 human pluripotent stem cells (hPSCs) in response to ACL treatment.

**Figure supplement 4—source data 1.** Fluorescence intensity and qPCR analysis of SHEF6 human pluripotent stem cells (hPSCs) in response to ACL.

**Figure supplement 5.** Flow cytometry gating strategy for isolating the SOX17::tdTomato+ cells within human extended pluripotent stem cell (hEPSC)-ACL, Rset-ACL, and human embryonic stem cell (hESC)-ACL populations.

comparisons in coefficients are only made across cell lines. This analysis revealed that all three cell populations bore a greater similarity to the hypoblast following ACL treatment, while their similarity to the epiblast decreased, suggesting a shift towards an endodermal fate (*Figure 1L*). Interestingly, Rset-ACL bore the greatest similarity of the three ACL-treated cell lines with the human hypoblast, while hESC-ACL was the most similar of the three to the definitive endoderm (*Figure 1L*).

## Pluripotent state determines the endodermal fate specified in hPSCs in response to ACL

EOMES, GATA6, and SOX17 are pan-endodermal markers, which are expressed in both extra-embryonic endoderm and embryonic definitive endoderm across mammalian models (*Stern and Downs, 2012*). To date, no putative markers for *in vivo* human extra-embryonic endoderm vs. definitive endoderm have been designated. We, therefore, sought to identify reliable human extra-embryonic endoderm markers that would allow the specific endodermal identity of the ACL-treated hPSCs to be determined. Previous studies have proposed markers for both human definitive and extra-embryonic endoderm based on the differentiation of hPSC towards extra-endodermal lineages (*Linneberg-Agerholm et al., 2019*) or have relied on putative mouse markers (*Murakami et al., 2004*; *Wang et al., 2012*). Contrastingly, we determined all of the differentially expressed genes between human extra-embryonic and definitive endoderm using published datasets of human embryos cultured *in vitro* (*Xiang et al., 2020*; *Zhou et al., 2019*; *Molè et al., 2021*) and hESC-derived definitive endoderm (*Cuomo et al., 2020*). This revealed that 24,894 of the 26,510 genes analysed were shared by both endodermal lineages, demonstrating considerable similarity between embryonic and extra-embryonic endoderm (*Figure 2A*). Nonetheless, 366 and 1250 genes were found to be differentially upregulated in the definitive endoderm and extra-embryonic endoderm, respectively (*Figure 2A* and *Supplementary file 2*). Extra-embryonic endoderm markers included *APOA1, APOA2, APOE,* and *APOC1* (*Figure 2B* and *Supplementary file 2*) in agreement with the function of the yolk sac as the primary site of apolipoprotein synthesis during early primate, chick and mouse development (*Baardman et al., 2013*; *Cindrova-Davies et al., 2017*). Similarly, *PDGFRα, NID2, FN1, HNF1β, HNF4a, COL4A1, PITX2,* and *PODXL* were also identified as human extra-embryonic endoderm markers (*Figure 2B* and *Supplementary file 2*), in line with the mouse extra-embryonic

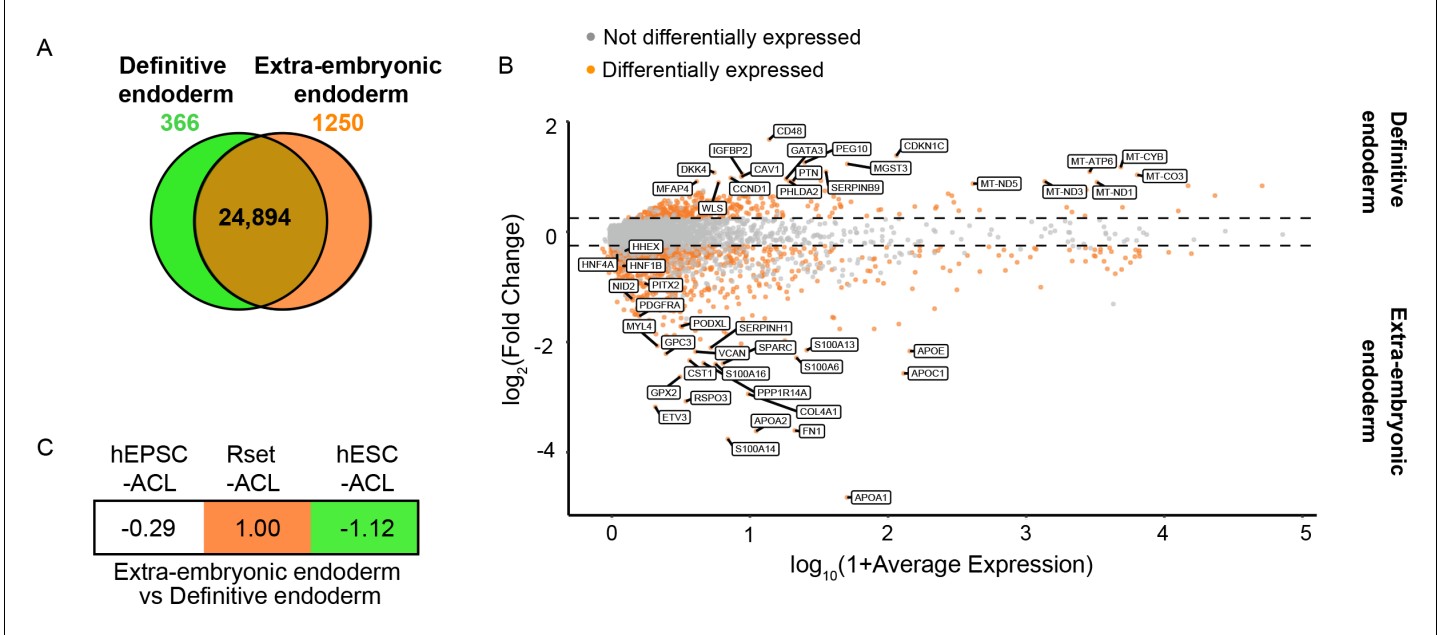

**Figure 2.** Transcriptional profiling of ACL-treated human pluripotent stem cells (hPSCs) relative to extra-embryonic endoderm and definitive endoderm expression signatures. (**A**) Venn diagram depicting the number of shared genes between human extra-embryonic endoderm (E6–14) and human embryonic stem cell (hESC)-derived definitive endoderm (brown), of extra-embryonic endoderm differentially expressed genes (orange), and of definitive endoderm differentially expressed genes (green). (**B**) MA plot representing the differentially expressed genes between human definitive endoderm and extra-embryonic endoderm. Top 20 differentially expressed genes are labelled, along with key extra-embryonic endoderm markers NID2, PDGFRa, HNF4a, HNF1β, PITX2, PODXL, and HHEX (>10% of cell type of interest, $\log_2$FC>0.25, p<0.05). (**C**) Transcriptomic-signature comparison score of human extra-embryonic endoderm vs. definitive endoderm within ACL-treated hPSCs (gene set variation analysis ([GSVA]) score for negative markers subtracted from GVSA score for positive markers and values normalised to 1). A negative value represents definitive endoderm similarity and a positive value represents an extra-embryonic endoderm similarity. hEPSC-ACL: n=2 technical replicates, Rset-ACL: n=3 technical replicates, hESC-ACL: n=2 technical replicates. Sc-RNA-seq expression data (extra-embryonic endoderm and definitive endoderm) and bulk RNA-seq expression data (cell lines) were used for all relevant panels.

endoderm expression profile at various stages of development (*Ghatpande et al., 2002*; *Johansson et al., 1997*; *Kwon et al., 2008*; *Rodriguez et al., 2005*; *Shahbazi et al., 2017*).

Using the makers identified above, we subjected ACL-treated hEPSC, Rset, and hESC populations to GSVA. Given that these marker sets are related – the negative markers for the extra-embryonic endoderm are the positive markers for the definitive endoderm – we then subtracted the GVSA marker score for the definitive endoderm from the GVSA marker score for extra-embryonic endoderm and normalised the highest score to 1. This allowed us to obtain a relative view of how similar each ACL cell type is to the extra-embryonic endoderm vs. the definitive endoderm. This revealed hESC-ACL to be the most similar to human definitive endoderm of the three populations, and Rset-ACL the most similar to extra-embryonic endoderm (*Figure 2C*). hEPSC-ACL were also more similar to definitive endoderm than extra-embryonic endoderm (*Figure 2C*).

Collectively, this transcriptional profiling of ACL-treated hPSCs indicates that the initial pluripotent state determines the endodermal fate that ACL treatment induces in hPSC populations. The fact that primed hESCs give rise to the most definitive endoderm-like population of the three is in line with previous findings (*Linneberg-Agerholm et al., 2019*). Although hEPSC-ACL are more similar to the definitive endoderm relative to extra-embryonic endoderm, they are not as similar to the definitive endoderm as hESC-ACL are, perhaps due to hEPSC-ACL representing a highly heterogenous population of cells. Rset-ACL were the most similar to the extra-embryonic endoderm, making them a potential candidate for modelling the human yolk sac.

## Developmental characterisation of ACL-treated hPSCs

We next wished to determine the dynamics of conversion from a pluripotent to an endoderm fate. Commitment to a definitive endoderm fate follows an initial intermediate mesendoderm fate

(*Kimelman and Griffin, 2000*), whereas extra-embryonic endoderm derives directly from the ICM without this intermediate state (*Niakan et al., 2012*). To determine whether a mesendoderm fate is induced during ACL treatment in hEPSCs, Rset hPSCs, or hESCs, we monitored the dynamics of SOX2, SOX17, and BRACHYURY during the 6 days of ACL treatment (*Figure 3A & B*). This revealed a trend of increase in BRACHYURY expression in hESC-ACL and hEPSC-ACL populations around day 2 of treatment, which then diminished by day 4 in both cases. However, in Rset hPSCs, this trend of BRACHYURY levels was not observed. This shows that of the three cell types, the specification dynamics of Rset hPSCs in response to ACL was the most similar to that expected of extra-embryonic endoderm specification.

To further profile ACL-treated cells, we generated human-mouse embryo chimeras. We aggregated eight-cell stage mouse embryos with small groups (three to six cells) of SOX17::H2B-tdTomato+ ACL-treated hEPSCs, Rset hPSCs, or hESCs. To ensure viability of ACL-treated cells throughout the chimeric process, embryos were cultured in ACL for 24 hr, before being transferred to mouse embryo medium (KSOM) for the remaining 24 hr until late blastocyst stage (*Figure 3C*). As a control we cultured eight-cell stage mouse embryos in ACL for 24 hr, before transferring them into KSOM and confirmed this had no effect on lineage specification and blastocyst formation (*Figure 3—figure supplement 1*). We observed that both hEPSC-ACL and Rset-ACL were able to contribute to the primitive endoderm of mouse embryos (*Figure 3D–F*). This was confirmed by the presence of reporter fluorescence within the ICM, and also by the co-expression of GATA6 with SOX17::H2B-tdTomato. hEPSC-ACLs and Rset-ACLs contributed to the mouse primitive endoderm in 18% and 12% of chimeras, respectively, with hEPSC-ACL contributing to both the epiblast and the primitive endoderm 3% of the time, suggesting a degree of fate plasticity (*Figure 3F*). This contribution profile of hEPSC-ACL and Rset-ACL contrasts that of hEPSCs and Rset hPSCs which both almost solely contribute to the epiblast compartment within human-mouse chimeric blastocysts: hEPSCs and Rset hPSCs contribute to the epiblast in 44% and 81% of embryos, and to both the epiblast and primitive endoderm 6% and 4% of the time, respectively (*Figure 3—figure supplement 2*). Primed hESCs treated with ACL could not contribute to mouse primitive endoderm (*Figure 3—figure supplement 3*).

Collectively, based on the transcriptional and functional evidence, we concluded that Rset-ACL are the most similar to human extra-embryonic endoderm relative to the other two cell lines. Despite hEPSC-ACL being able to contribute to the murine primitive endoderm, and thereby sharing some functional properties with the extra-embryonic endoderm, they also share significant similarities to the definitive endoderm, both in terms of their developmental pathway and transcriptional profile. This is likely due to the high degree of heterogeneity present in hEPSC-ACL cultures. hESC-ACL are more similar to definitive endoderm, functionally, transcriptionally, and in terms of their developmental trajectory. Based on these findings we concluded that Rset-ACL were the most promising candidate for modelling the human yolk sac.

## Pluripotent state determines the developmental state of extra-embryonic endoderm specified in hPSCs in response to ACL

To model the yolk sac-epiblast crosstalk, we sought to use an extra-embryonic endoderm cell type that resembles the post-implantation hypoblast-derived yolk sac. Therefore, we next ascertained whether Rset-ACL were more similar to pre- or post-implantation extra-embryonic endoderm. However, to date no markers have been defined that distinguish the two human extra-embryonic endoderm states. Following comparison between the transcriptional profile of pre-implantation extra-embryonic endoderm (primitive endoderm) (E6–7) (*Xiang et al., 2020*; *Zhou et al., 2019*) and post-implantation extra-embryonic endoderm (yolk sac) (E10–14) (*Molè et al., 2021*; *Xiang et al., 2020*; *Zhou et al., 2019*), we identified differentially expressed genes between the two states. Of the 26,510 genes examined, 25,496 were shared, with 397 and 617 genes being differentially upregulated in the pre- or post-implantation state, respectively (*Figure 4A*) (*Supplementary file 3*). Interestingly, *AXL*, a visceral endoderm marker in the mouse (*McDonald et al., 2014*), was among the identified markers of post-implantation yolk sac (*Figure 4B* and *Supplementary file 3*). The presence of *APOA2* and *APOB* in the top differentially expressed post-implantation genes once again alludes to the yolk sac as the primary site of apolipoprotein synthesis in the early embryo.

We then analysed the expression of the pre- and post-implantation markers in Rset-ACL, and also the only previously reported hPSC-derived extra-embryonic endoderm progenitor cell line called

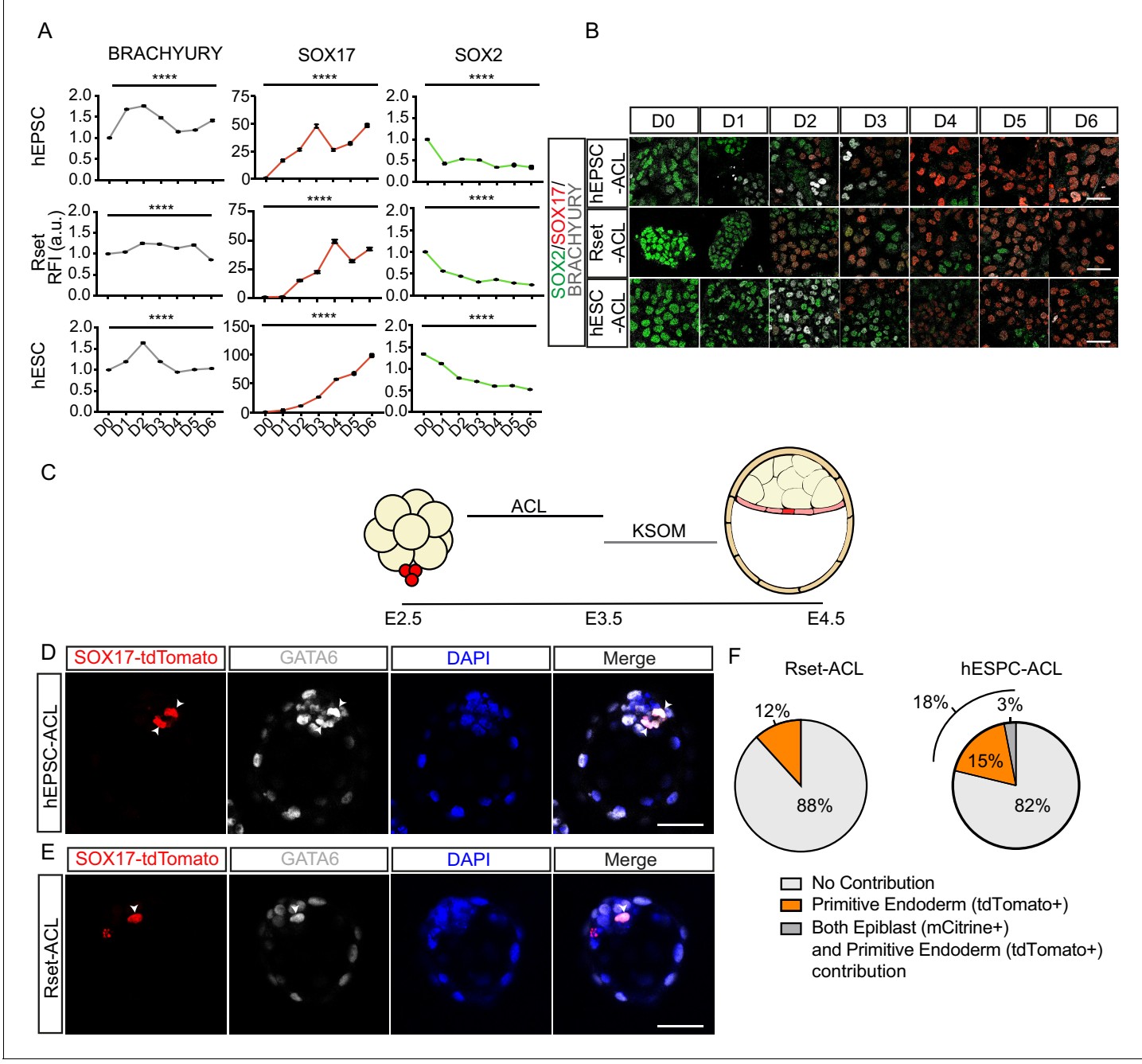

**Figure 3.** Functional characterisation of ACL-treated human pluripotent stem cells (hPSCs). (**A**) Line graph (± SEM) plotting the relative fluorescence intensity (RFI arbitrary units [a.u.]) of SOX2, SOX17, and BRACHYURY in ACL-treated population. Kruskal–Wallis test, ****p < 0.0001 (number of cells analysed SOX17: human extended pluripotent stem cell (hEPSC): D0 n=2774, D1 n=3258, D2 n = 2932, D3 n=3292, D4 n=2459, D5 n=2293, D6 n=1883, Rset: D0 n=2108, D1 n=3580, D2 n=2738, D3 n=2586, D4 n=2149, D5 n=2698, D6 n = 2441, human embryonic stem cell (hESC): D0 n=1426, D1 n=1924, D2 n=1943, D3 n=2817, D4 n=2248, D5 n=1148, D6 n=1957 (n=2 independent experiments); SOX2: hEPSC: D0 n = 2774, D1 n=3258, D2 n=2932, D3 n=3292, D4 n=2459, D5 n=2293, D6 n=1883, Rset: D0 n=2342, D1 n=3634, D2 n=2794, D3 n=2671, D4 n=2068, D5 n=2293, D6 n=2918, hESC: D0 n=2353, D1 n=3082, D2 n=3129, D3 n=4174, D4 n=2989, D5 n=1787, D6 n=3114; BRACHYURY: hEPSC: D0 n = 3452, D1 n=4090, D2 n=3435, D3 n=4223, D4 n=2881, D5 n=2837, D6 n=2218, Rset: D0 n=2317, D1 n = 3739, D2 n=3141, D3 n=2881, D4 n=2467, D5 n=2951, D6 n=4400, hESC: D0 n=2685, D1 n=3433, D2 n=3997, D3 n=5325, D4 n=3490, D5 n=2155, D6 n = 3258. n = 3 independent experiments). (**B**) Immunofluorescence images of SOX2 (green), SOX17 (red), and BRACHYURY (grey) in hEPSCs, Rset hPSCs, and hESCs during 6 days of ACL treatment (scale bar = 50 μm). (**C**) Schematic of human-mouse chimera protocol using ACL-treated cells. Clumps of three to six SOX-17::H2B-tdTomato+ treated cells were aggregated with E2.5 mouse embryos and cultured for 24 hr in ACL. After this, embryos were moved into KSOM and cultured for another 24 hr. (**D**) Immunofluorescence images of SOX17::H2B-tdTomato (red), GATA6 (grey), and DAPI (blue) in E4.5 mouse blastocysts. hEPSC-ACL cells contribution

*Figure 3 continued on next page*

*Figure 3 continued*

denoted by SOX17::H2B-tdTomato+ (red) cells (white arrowheads) (scale bar = 50 µm). (E) Immunofluorescence images of SOX17::H2B-tdTomato (red), GATA6 (grey), and DAPI (blue) in E4.5 mouse blastocysts. Rset-ACL cells contribution denoted by SOX17::H2B-tdTomato+ (red) cells (white arrowheads). (F) Pie charts showing percentage of human cell contribution to the primitive endoderm, epiblast, or both. Rset (N = 67 embryos, n=3 independent experiments), hEPSC (N = 39 embryos, n=3 independent experiments).

The online version of this article includes the following source data and figure supplement(s) for figure 3:

**Source data 1.** Fluorescence intensity analysis of ACL-treated human pluripotent stem cells (hPSCs) and analysis of mouse-human pre-implantation chimeras.

**Figure supplement 1.** Characterisation of mouse blastocysts cultured in interspecies chimera conditions.

**Figure supplement 1—source data 1.** Analysis of mouse embryo culture condition effects and mouse-human chimeras.

**Figure supplement 2.** Contribution of human extended pluripotent stem cell (hEPSC) and Rset human pluripotent stem cells (hPSCs) to interspecies chimeras.

**Figure supplement 2—source data 1.** Analysis of mouse-human blastocyst chimeras using human extended pluripotent stem cells (hEPSCs) and Rset human pluripotent stem cells (hPSCs).

**Figure supplement 3.** Contribution of ACL-treated human embryonic stem cells (hESCs) to interspecies chimeras.

**Figure supplement 3—source data 1.** Analysis of mouse-human blastocyst chimeras using human embryonic stem cell (hESC)-ACL.

T2iLGö-PrE (*Linneberg-Agerholm et al., 2019*) derived from T2iLGö hPSCs which harbour a pluripotent state comparable to the pre-implantation naïve human epiblast (*Guo et al., 2017*). We postulated that given T2iLGö-PrE were derived from an hPSC representing an earlier developmental epiblast state than Rset hPSCs, they may be more similar to the pre-implantation extra-embryonic

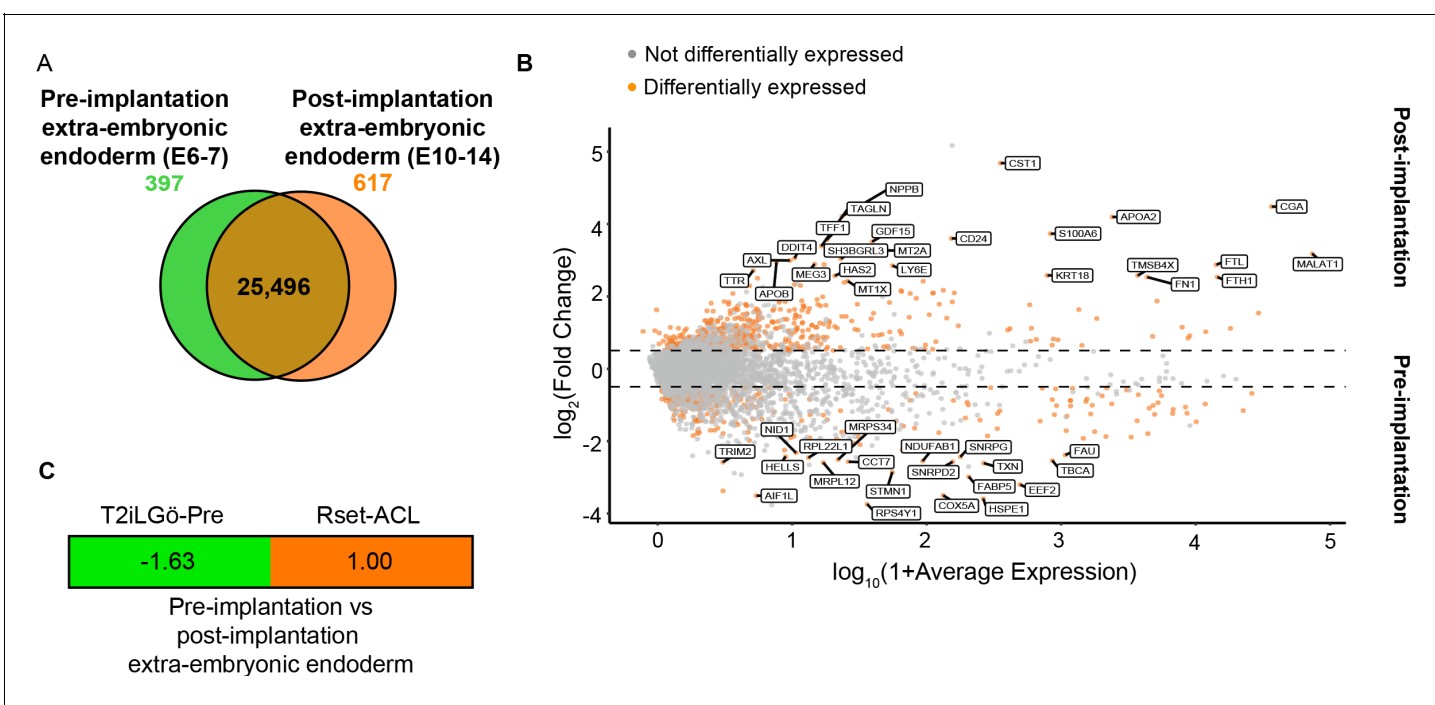

**Figure 4.** Transcriptional profiling of T2iLGö-PrE and Rset-ACL relative to human pre- and post-implantation extra-embryonic endoderm expression signatures. (A) Venn diagram depicting the number of shared genes between pre-implantation extra-embryonic endoderm (E6–7) and post-implantation extra-embryonic endoderm (E10–14) (brown), of pre-implantation extra-embryonic endoderm differentially expressed genes (green) and of post-implantation extra-embryonic endoderm differentially expressed genes (orange). (B) MA plot representing the differentially expressed genes between human pre-implantation and post-implantation extra-embryonic endoderm. Top 20 differentially expressed genes are labelled (>10% of cell type of interest, log$_2$FC>0.25, p<0.05). (C) Transcriptomic-signature comparison score of pre-implantation extra-embryonic endoderm vs. post-implantation extra-embryonic endoderm for T2iLGö-Pre and Rset-ACL (gene set variation analysis [GSVA] score for pre-implantation extra-embryonic endoderm subtracted from GSVA score for post-implantation extra-embryonic endoderm and values normalised to 1). A negative value (green) represents pre-implantation extra-embryonic endoderm similarity and a positive value (orange) represents post-implantation extra-embryonic endoderm. T2iLGö-Pre: n=2 technical replicates (from published dataset), Rset-ACL: n=3 technical replicates. Sc-RNA-seq expression data (pre- and post-implantation extra-embryonic endoderm) and bulk RNA-seq expression data (cell lines) were used for all relevant panels.

endoderm (primitive endoderm), whereas Rset-ACL may be more comparable to the post-implantation yolk sac.

Similar to the process described above, we performed a differential gene expression analysis between the pre-implantation (E6–7) (*Xiang et al., 2020*; *Zhou et al., 2019*) and post-implantation (E10–14) (*Molè et al., 2021*; *Xiang et al., 2020*; *Zhou et al., 2019*) extra-embryonic markers on the two cell lines to determine which was more similar to the post-implantation yolk sac. Given that these marker sets are related – the negative markers for the post-implantation extra-embryonic are the positive markers for the pre-implantation extra-embryonic endoderm – we then subtracted the GSVA marker score for the pre-implantation markers from the GSVSA marker score for post-implantation markers and normalised the highest score to 1. This analysis revealed Rset-ACL to be the most similar to the post-implantation extra-embryonic endoderm, while T2iLGö-PrE were more similar to the pre-implantation extra-embryonic endoderm (*Figure 4C*). These results indicate that Rset-ACL is the best candidate for modelling yolk sac-epiblast interactions. Accordingly, these cells will be referred to as YSLCs.

## YSLCs antagonise WNT and BMP signalling in hESCs

The yolk sac of the mouse embryo harbours an asymmetrically localised subpopulation of cells called the AVE. These cells secrete inhibitors of BMP, WNT, and NODAL signalling, thereby creating a signalling gradient across the post-implantation epiblast. This gradient in turn determines the positioning of the primitive streak and, therefore, the anterior-posterior axis of the embryo (*Arnold and Robertson, 2009*). The post-implantation hypoblast of both the cynomolgus monkey and human embryo also expresses *DKK1*, *CER1*, and *NOG* (*Molè et al., 2021*; *Nakamura et al., 2016*; *Sasaki et al., 2016*) and studies of human embryos cultured to post-implantation stages *in vitro* have suggested that an anterior subpopulation of cells within the human hypoblast may influence BMP signalling within the adjacent epiblast (*Molè et al., 2021*). This implies that the human yolk sac may harbour AVE-like cells. We, therefore, analysed the expression of these pathway inhibitors within the hypoblast of the pre- and post-implantation human embryo using published datasets (*Molè et al., 2021*; *Xiang et al., 2020*; *Zhou et al., 2019*) and found that *DKK1, CER1, LEFTY2, and SFRP1* are expressed at greater levels in post-implantation relative to pre-implantation hypoblast, while *NOG* is expressed at higher levels in the pre-implantation hypoblast (*Figure 5A*).

Given that YSLCs share transcriptional similarity to the post-implantation extra-embryonic endoderm, we examined the expression of WNT and BMP inhibitors in YSLCs and observed that all of the analysed inhibitors were upregulated in YSLCs (*Figure 5B* and *Figure 5—figure supplement 1A and B*). In human stem cell models, DKK1 and NOG act as WNT and BMP inhibitors, respectively (*Martyn et al., 2019*; *Simunovic et al., 2019*), whereas CER1 antagonises both NODAL and BMP signalling in other systems (*Aykul and Martinez-Hackert, 2016*). We, therefore, postulated that YSLCs may share functional characteristics with the mouse AVE and the human anterior hypoblast in their ability to govern cell fate via BMP and WNT inhibition.

To test this, we first sought to determine whether YSLCs can influence hESC fate. hESCs have been shown to lose their pluripotent character in a BMP-dependent manner when cultured in a 3D 'sandwich' matrix of Geltrex, in which cells are plated on top of a base layer of Geltrex before being covered by mTESR1 containing 5% Geltrex (*Shao et al., 2017b*). Confirming these previous findings, we found that at a density of 60,000 cells per cm$^2$ hESCs lost SOX2 expression and either formed squamous spheroids or lost structural integrity (*Figure 5—figure supplement 1C*). Similarly, we found that inhibiting WNT signalling using either IWR1 (10 ng ml$^{-1}$) or XAV (10 ng ml$^{-1}$) in hESCs cultured in 3D Geltrex prevented downregulation of SOX2 (*Figure 5—figure supplement 1C and D*). Collectively, this indicates that hESCs lose SOX2 in 3D Geltrex in response to BMP and WNT signalling pathway activation. To test whether YSLCs could antagonise this signalling, YSLCs were cultured at a 1:1 ratio with hESCs for 72 hr in mTeSR1 PLUS, revealing SOX2 levels to be higher in co-culture conditions relative to control hESCs. We also noted a slight (but not significant) increase in the number of epithelial spheroids (squamous and columnar) relative to disorganised structures in co-culture conditions (*Figure 5—figure supplement 1E–G*). This demonstrates an ability of YSLCs to inhibit the loss of SOX2 in hESCs that would otherwise occur in 3D Geltrex.

To test whether the reduced loss of SOX2 expression was controlled by a soluble factor, we conditioned mTeSR1 PLUS medium for 24 hr on confluent YSLCs. This experimental setting also removed any effects of suboptimal YSLC viability within the co-culture conditions. hESCs were then

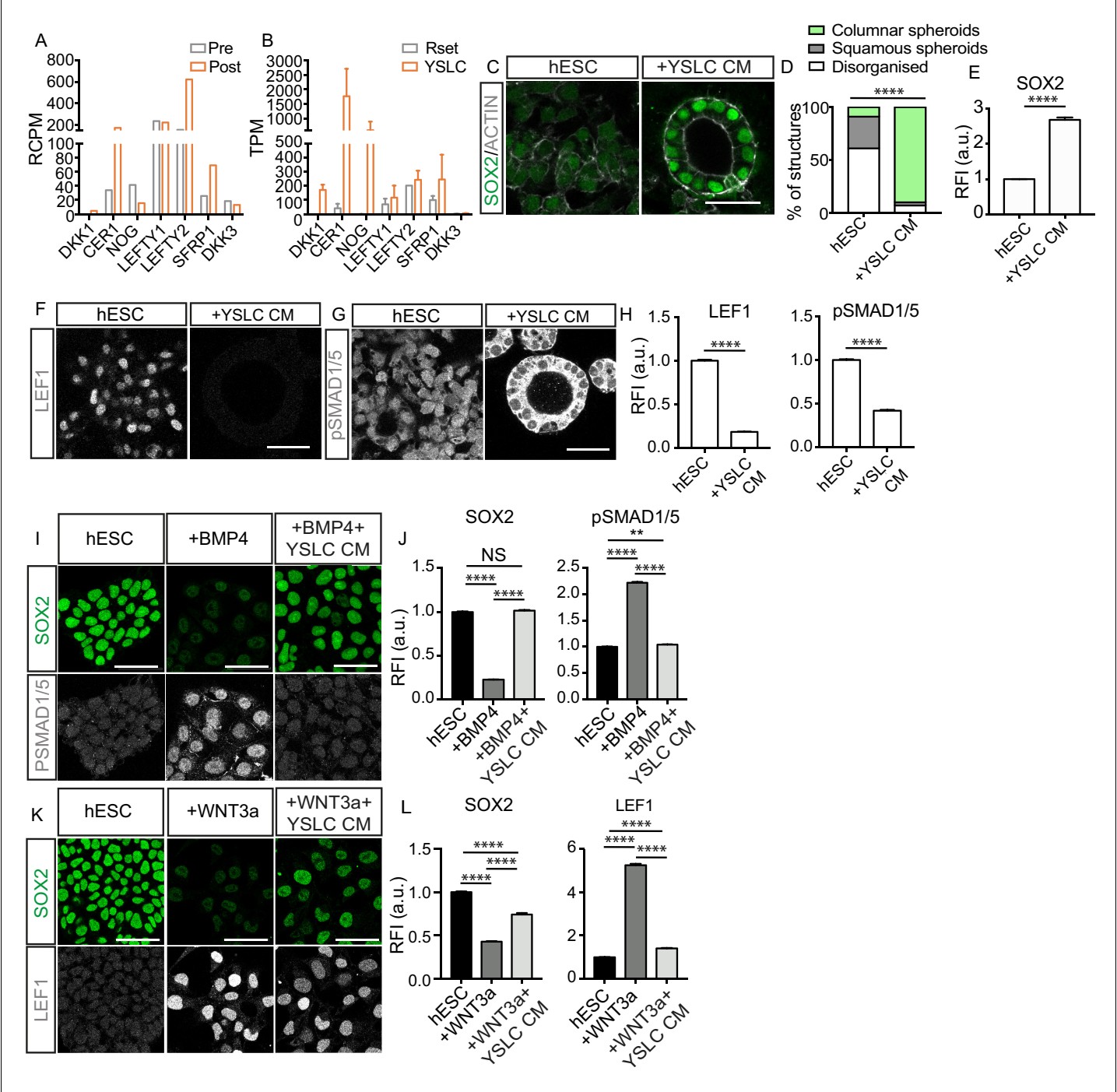

**Figure 5.** Yolk sac-like cells (YSLCs) antagonise BMP and WNT signalling in human embryonic stem cells (hESCs). (A) Bar chart plotting average expression levels of WNT, NODAL, and BMP inhibitors in pre-implantation (Pre, E6–7) and post-implantation human hypoblast (Post, E10–14) denoted as relative counts per million (RCPM) using published sc-RNA-seq data. (B) Bar chart (± SEM) plotting expression levels of BMP, NODAL, and WNT inhibitors in Rset hPSCs and YSLCs denoted as transcripts per million (TPM) using bulk RNA-seq data (YSLCs: n=3 technical replicates, Rset: n=2 technical replicates [from published dataset]). (C) Immunofluorescence images of SOX2 (green) and ACTIN (grey) in hESCs cultured in 3D Geltrex in either mTeSR (hESC) or mTeSR conditioned on YSLCs (+YSLC CM) (scale bar = 50 μm). (D) Percentage of structures that are columnar, squamous, or disorganised based on cell aspect ratio calculations in cells from panel C using ACTIN stainings. Chi-squared test. ****p < 0.0001 (number of structures analysed: control n=33, +YSLC CM n=30, n=3 independent experiments each with one sample). (E) Bar chart (± SEM) depicting SOX2 levels as inferred from relative fluorescence intensity (RFI arbitrary units [a.u.]) in cells from panel C. Mann–Whitney U-test; ****p < 0.0001 (number of cells analysed: SOX2: hESC n=2346, +YSLC CM n=596, n=3 independent experiments each with one sample). (F) Immunofluorescence of pSMAD1/5 (grey) in hESCs cultured in mTeSR (hESC) or mTeSR conditioned on YSLCs (+YSLC CM) in 3D Geltrex (scale bar = 50 μm). (G) Immunofluorescence of LEF1 (grey) in

*Figure 5 continued on next page*

*Figure 5 continued*

hESCs cultured in mTeSR (hESC) or mTeSR conditioned on YSLCs (+YSLC CM) in 3D Geltrex (scale bar = 50 μm). (H) Bar chart (± SEM) depicting the levels of pSMAD1/5 (nuclear) and LEF1 in hESCs cultured in mTeSR (hESC) or mTeSR conditioned on YSLCs (+YSLC CM) as inferred from RFI in cells from panels F and G, respectively. Mann–Whitney U-test, ****p <0.0001 (number of cells analysed: pSMAD1/5: hESC n=2563, +YSLC CM n=1660; LEF1: hESC n=1984, +YSLC CM n=1041. n=3 independent experiments each with one sample). (I) Immunofluorescence of SOX2 (green) and pSMAD1/5 (grey) in hESCs cultured in mTeSR (hESC), treated with BMP4 (+BMP4) or treated with BMP4 and YSLC conditioned media (+BMP4+YSLC CM) (scale bar = 50 μm). (J) Bar charts (± SEM) depicting the levels of SOX2 and pSMAD1/5 RFI in samples from panel I. Kruskal–Wallis test with Dunn's multiple comparisons test, ****p<0.0001 (number of cells analysed: hESC n=1549, +BMP4 n = 1063, +BMP4+YSLC CM n=1544, n=3 independent experiments each with one sample). (K) Immunofluorescence of SOX2 (green) and LEF1 (grey) in hESCs cultured in mTeSR (hESC), treated with WNT3a (+WNT3a) or treated with WNT3a and YSLC conditioned media (+WNT3a+YSLC CM) (scale bar = 50 μm). (L) Bar chart (± SEM) depicting the levels of SOX2 and LEF1 RFI in samples from panel K. Kruskal–Wallis test with Dunn's multiple comparisons test, ****p<0.0001 (number of cells analysed: hESC n=1604, +WNT3a n=1682, +WNT3a+YSLC CM n=1376, n=3 independent experiments each with one sample).

The online version of this article includes the following source data and figure supplement(s) for figure 5:

**Source data 1.** Analysis of BMP/WNT/NODAL inhibitors in the human embryo and yolk sac-like cells (YSLCs) and analysis of human embryonic stem cell (hESC) response to YSLCs.

**Figure supplement 1.** Yolk sac-like cells (YSLCs) prevent SOX2 and NANOG loss.

**Figure supplement 1—source data 1.** Fluorescence intensity analysis of SOX2 and NANOG levels in human embryonic stem cells (hESCs) in the presence of yolk sac-like cells (YSLCs) and YSLC conditioned media (CM).

cultured in this conditioned medium for 72 hr in 3D Geltrex. mTeSR1 PLUS conditioned on YSLCs blocked the downregulation of SOX2 within hESCs relative to control conditions and also significantly supported a columnar epithelial morphology (*Figure 5C–E*). These results show that YSLCs are able to prevent SOX2 downregulation and support columnar spheroid formation in hESCs in 3D Geltrex via secreted components, but that the potency of this effect is limited when they are physically co-cultured with hESCs, likely due to a reduced viability of YSLCs in 3D Geltrex conditions.

To understand whether YSLC conditioned medium was inhibiting BMP and WNT signalling within the 3D hESC spheroids, we analysed the levels of the downstream WNT and BMP effectors LEF1 and pSMAD1/5, respectively, in hESCs cultured in 3D Geltrex in either mTeSR1 PLUS or mTeSR1 PLUS conditioned on YSLCs. This revealed that both pathways were active in mTeSR1 PLUS control conditions, but their activity decreased in the presence of YSLC conditioned medium (*Figure 5F–H*).

To more precisely control the signalling pathways inducing SOX2 loss in hESCs, we added either BMP4 or WNT3a to hESCs cultured in 2D. Addition of 1 ng ml$^{-1}$ of BMP4 to mTeSR1 PLUS led to a marked decrease in SOX2 expression concomitant with an increase in the levels of nuclear pSMAD1/5 (*Figure 5I,J*). Strikingly, when hESCs were exposed to BMP4 in the presence of YSLC conditioned medium, SOX2 expression did not significantly decrease and levels of nuclear pSMAD1/5 remained low (*Figure 5I,J*). Similarly, when hESCs were cultured in the presence of 1 μg ml$^{-1}$ WNT3A they lost SOX2 expression and upregulated LEF1 (*Figure 5K,L*). However, when hESCs were cultured in the presence of 1 μg ml$^{-1}$ WNT3a and YSLC conditioned medium, the downregulation of SOX2 and upregulation of LEF1 was significantly counteracted (*Figure 5K–L*). A similar trend was observed for NANOG levels in hESCs that were exposed to either BMP4 or WNT3a in the presence of YSLC conditioned medium (*Figure 5—figure supplement 1H–K*). Together, our results demonstrate that YSLCs block loss of pluripotent gene expression by antagonising WNT and BMP signalling via secreted factors and can thereby influence hESC fate.

## YSLCs inhibit posterior epiblast fate specification in hESCs

Within the mouse embryo, the AVE acts to induce anterior fate and repress the formation of the primitive streak in the epiblast cells adjacent to it (*Rivera-Pérez and Hadjantonakis, 2015*). A similar function has been recently proposed for the human anterior hypoblast (*Molè et al., 2021*). Given the ability of YSLCs to influence hESC fate via WNT and BMP inhibition, we hypothesised that YSLCs may bear similarity to both the mouse AVE and the human anterior hypoblast, and may, therefore, be able to prevent hESCs from differentiating into posterior epiblast fates. Consequently, we analysed the expression of the posterior markers BRACHYURY and GATA6 in hESCs cultured in 3D Geltrex. In control conditions, BRACHYURY was not induced, while GATA6 was upregulated, and this upregulation was blocked in the presence of YSLC conditioned medium (*Figure 6A–C*). Next, to test whether YSLCs can block mesoderm specification, we employed a 2D mesoderm induction protocol

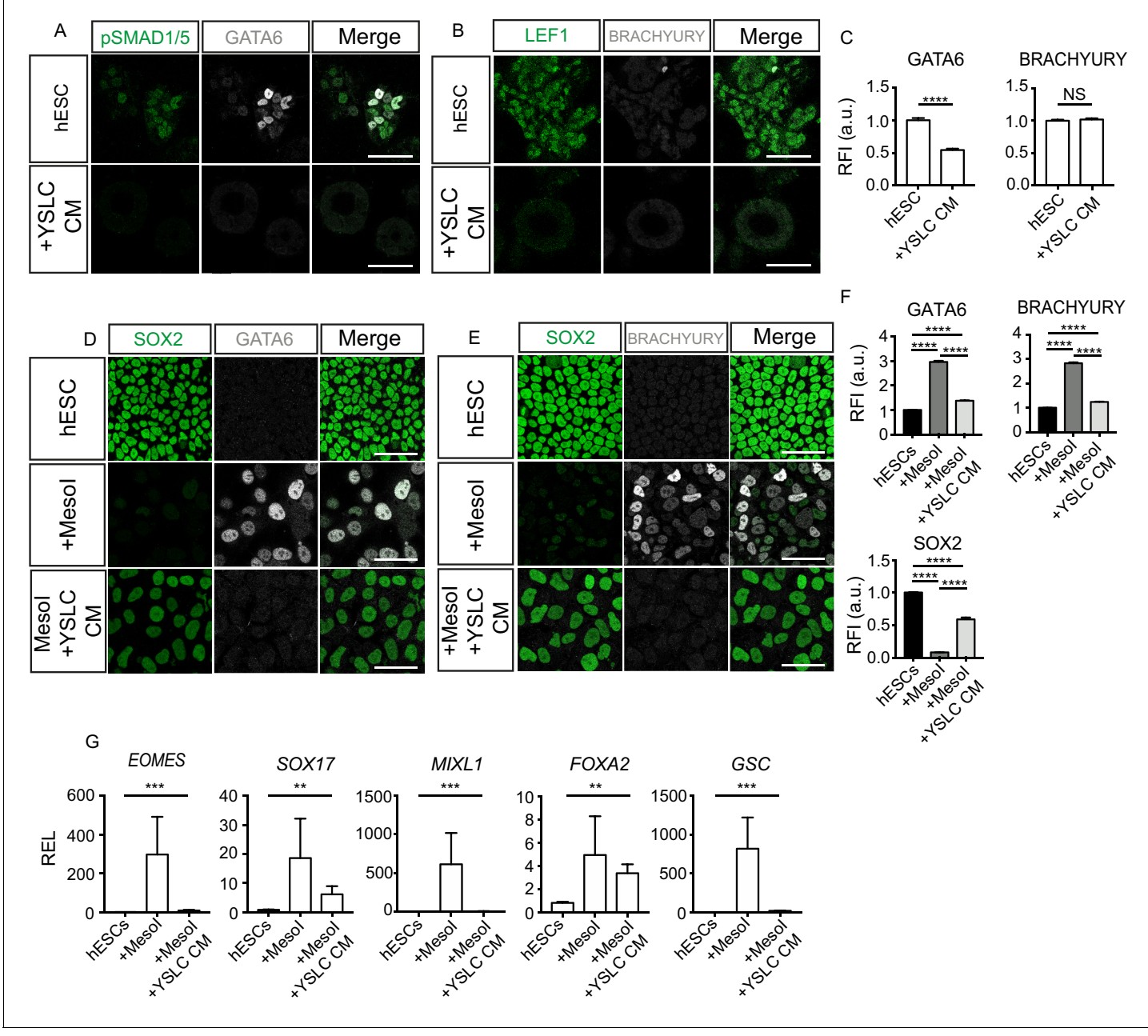

**Figure 6.** Yolk sac-like cells (YSLCs) inhibit mesoderm and endoderm specification in human embryonic stem cells (hESCs) in 3D and 2D. (**A**) Immunofluorescence of pSMAD1/5 (green) and GATA6 (grey), and a merge of the two, in hESCs cultured in 3D Geltrex in mTeSR (hESC) or YSLC conditioned media (+YSLC CM) (scale bar = 50 μm). (**B**) Immunofluorescence of LEF1 (green) and BRACHYURY (grey), and a merge of the two, in hESCs cultured in 3D Geltrex in mTeSR (hESC) or YSLC conditioned media (+YSLC CM) (scale bar = 50 μm). (**C**) Bar charts (± SEM) depicting the levels of GATA6 and BRACHYURY in hESCs cultured in 3D Geltrex in mTeSR (hESC) or YSLC conditioned media (+YSLC CM) as inferred from relative fluorescence intensity (RFI) arbitrary units [a.u.] in cells from panels A and B, respectively (number of cells analysed: GATA6: hESC n=2336, YSLC CM n=1074, BRACHURY: hESC n=2112, YSLC CM n=1261, n=3 independent experiments each with one sample) Mann–Whitney U-test; *p < 0.05, **p < 0.01, ***p < 0.001, ****p < 0.0001. (**D**) Immunofluorescence of SOX2 (green) and GATA6 (grey), and a merge of the two in hESCs, hESCs in MesoI (mesoderm induction medium), or MesoI + YSLC CM (scale bar = 50 μm). (**E**) Immunofluorescence of SOX2 (green) and BRACHYURY (grey), and a merge of the two, in hESCs, hESCs in mesoderm induction medium (MesoI), or MesoI conditioned on YSLCs (MesoI + YSLC CM) (scale bar = 50 μm). (**F**) Bar chart (± SEM) depicting the levels of GATA6, BRACHYURY and SOX2 in hESCs cultured in basal medium (hESC), hESCs in mesoderm induction medium (MesoI), or MesoI + YSLC CM as measured by RFI. Kruskal–Wallis test with Dunn's multiple comparisons test. ****p <0.0001 (number of cells analysed: GATA6: hESC n = 5583, +MesoI n=2554, +MesoI+YSLC CM n=2383, BRACHYURY: hESC n = 6859, +MesoI n=6741, +MesoI+YSLC CM n=2072; SOX2: hESC n=6859, +MesoI n=6741, +MesoI+ YSLC CM n=2072. n=3 independent experiments each with one sample). (**G**) Relative expression levels (REL) (± SEM) of *EOMES, SOX17, MIXL1, FOXA2,* and *GSC* in hESCs cultured in basal medium (hESC), hESCs in mesoderm induction

*Figure 6 continued on next page*

*Figure 6 continued*

medium (MesoI), or MesoI + YSLC CM as measured by RFI, normalised to their hESC control (n=4 samples, two independent experiments). Kruskal–Wallis test; *p < 0.05, **p < 0.01, ***p < 0.001, ****p < 0.0001.

The online version of this article includes the following source data for figure 6:

**Source data 1.** Fluorescence intensity and qPCR analysis of mesoderm and endoderm markers in human embryonic stem cells (hESCs) in the presence of yolk sac-like cell (YSLC) conditioned medium (CM).

involving the supplementation of 10 ng μl$^{-1}$ BMP4 to hESCs for 3 days (*Richter et al., 2014*). Strikingly, immunofluorescence analyses revealed that YSLC conditioned medium suppressed the upregulation of BRACHYURY while also preventing the loss of SOX2 in hESCs cultured in mesoderm induction medium (*Figure 6D–F*). GATA6 was also upregulated in hESCs after BMP4 treatment and this was also countered by YSLC conditioned medium. Similarly, YSLC conditioned medium suppressed the upregulation of the posterior markers *EOMES, SOX17, MIXL1, FOXA2,* and *GSC* at the RNA level (*Figure 6G*). These results, therefore, demonstrate that YSLCs are able to inhibit posterior epiblast marker expression in hESCs.

## YSLCs upregulate pluripotency and anterior ectoderm markers in hESCs

SOX2 is a pluripotency marker but is also expressed in the anterior ectoderm. Given that YSLCs are able to prevent SOX2 downregulation in 3D Geltrex, we investigated the levels of the pluripotency markers OCT4 and NANOG, and the anterior ectoderm markers SOX1 and OTX2 in hESC cultured in YSLC conditioned medium in 3D Geltrex over a 5-day period. This revealed that expression of both OCT4 and NANOG was higher in hESCs cultured in YSLC conditioned medium at all timepoints relative to control mTeSR conditions, while SOX2 levels were significantly different from day 3 of culture (*Figure 7A and B*). SOX1 levels steadily increased relative to controls over the 5-day culture period but were downregulated after 7 days (*Figure 7C and D*, *Figure 7—figure supplement 1A and B*). Conversely, OTX2 expression did not increase until day 3, and peaked at day 4, before decreasing again (*Figure 7C and D*). Interestingly, in the presence of conditioned medium, pluripotency markers and anterior ectoderm markers were co-expressed (*Figure 7—figure supplement 1C*). Overall, these results demonstrate that YSLCs can upregulate pluripotency and anterior ectoderm markers in hESCs in 3D.

To further assess the ability of YSLCs to induce anterior ectoderm markers in hESCs, we determined whether YSLC conditioned medium could upregulate neural markers in 2D. In neural progenitor differentiation protocols, dual small molecule targeting of SMAD signalling is widely used (*Chambers et al., 2009*; *Shi et al., 2012*). Given that YSLC conditioned medium was able to inhibit SMAD activity in hESCs in 2D and 3D, we analysed the ability of YSLC conditioned medium to replace small molecule SMAD inhibitors within a 2D neural differentiation protocol. Accordingly, hESCs were cultured in YSLC conditioned medium, rather than SB431542 and NOGGIN, for 11 days, before being cultured in FGF2 for 4 days (*Shi et al., 2012*; *Figure 7E*). After the completion of the 15-day protocol, the anterior ectoderm markers SOX1, *PAX6, NESTIN,* and *NOTCH1* were all significantly upregulated in hESCs (*Figure 7F–H*). SOX2 was downregulated relative from controls, but was still present, as has been observed during the early stages of neural specification (*Zhang et al., 2018*). Interestingly, YSLC conditioned medium induced the upregulation of anterior ectoderm markers to intermediate levels relative to hESCs cultured in dual SMAD conditions and control hESCs (*Figure 7—figure supplement 1D*). This was also the case for levels of pluripotency markers (*Figure 7—figure supplement 1D*). Overall these results demonstrate that YSLCs support anterior ectoderm marker expression in combination with pluripotency markers and, therefore, share functional similarities with the mammalian AVE.

## Discussion

The signalling pathways that govern lineage specification and morphogenesis within the implanting human embryo remain poorly understood due to technical and ethical constraints. Here, we have created an *in vitro* system to explore the signalling interactions between the epiblast and yolk sac of the post-implantation human embryo.

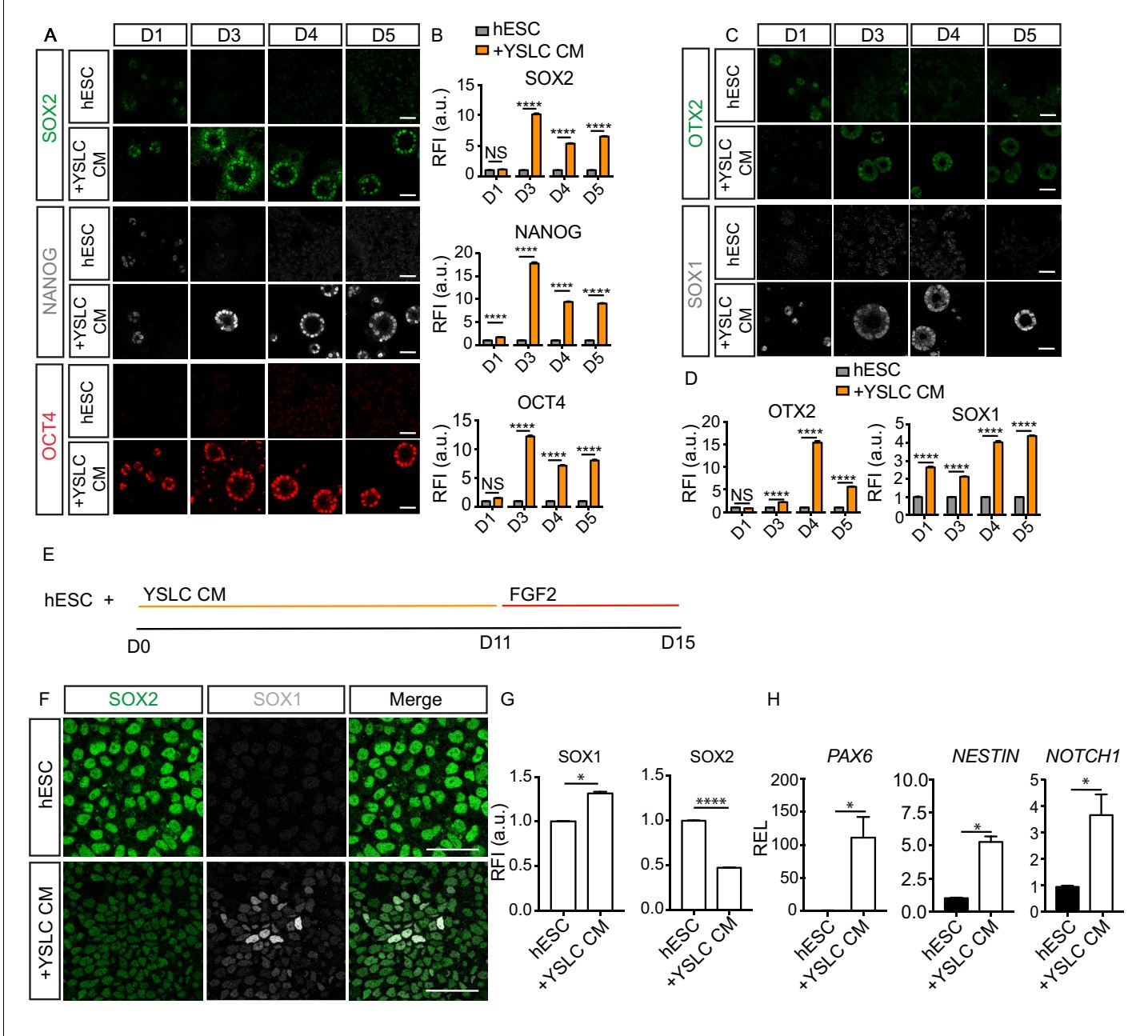

**Figure 7.** Yolk sac-like cells (YSLCs) induce the co-expression of anterior ectoderm markers and pluripotency markers in human embryonic stem cell (hESC) in 3D and 2D. (**A**) Immunofluorescence of SOX2 (green), NANOG (grey), and OCT4 (red) in hESCs after 1, 3, 4, or 5 days cultured in mTeSR (hESC) or in YSLC conditioned media (+YSLC CM) in 3D Geltrex (scale bar = 50 μm). (**B**) Bar charts (± SEM) depicting relative fluorescence intensity (RFI arbitrary units [a.u.]) of SOX2, NANOG, and OCT4 in hESCs cultured in YSLC CM (+YSLC CM) relative to hESCs cultured in mTeSR (hESCs) in 3D Geltrex for 1, 3, 4, and 5 days. Two-way ANOVA with Sidak's multiple comparisons test comparing hESC vs. +YSLC CM for each day. ****p<0.0001, ** <0.01, NS p>0.05 (number of cells analysed: SOX2: day 1: hESC n=567, +YSLC CM n=662, day 3: hESC n=2318, +YSLC CM n=1870, day 4: hESC n=3755, +YSLC CM n=1249, day 5: hESC n=4946, +YSLC CM n=1897, OCT4: day 1: hESC n=450, +YSLC CM n=601, day 3: hESC=1929, +YSLC CM n=1902, day 4: hESC=3472, +YSLC CM n=1314, day 5: hESC n=4946, +YSLC CM n=1897, NANOG: day 1: hESC n=450, +YSLC CM n=601, day 3: hESC n=1929, +YSLC CM n=1902, day 4: hESC n=3472, +YSLC CM n=1314, day 5: hESC n=4946, +YSLC CM n=1897, n = 3 independent experiments each with one sample). (**C**) Immunofluorescence of OTX2 (green) and SOX1 (grey) in hESCs after 1, 3, 4, or 5 days cultured in mTeSR (hESC) or YSLC conditioned media (+YSLC CM) in 3D Geltrex (scale bar = 50 μm). (**D**) Bar charts (± SEM) depicting RFI of OTX2 and SOX1 in hESCs cultured in YSLC CM (+YSLC CM) relative to hESCs cultured in mTeSR (hESC) in 3D Geltrex for 1, 3, 4, and 5 days. Two-way ANOVA with Sidak's multiple comparisons test comparing hESC vs. +YSLC CM for each day. ****p<0.0001 NS, p>0.05 (number of cells analysed: OTX2: day 1: hESC n=456, YSLC CM n=914, day 3: hESC n=4278, +YSLC CM n=2036, day 4: hESC n=2432, +YSLC CM n=1123, day 5: hESC n=3858, +YSLC CM n=3243, SOX1: day 1: hESC n=503,

*Figure 7 continued on next page*

*Figure 7 continued*

+YSLC CM n=735, day 3: hESC n=2933, +YSLC CM n=3023, day 4: hESC n=3426, +YSLC CM n=2507, day 5: hESC n=3764, +YSLC CM n=2620, n = 3 independent experiments each with one sample). (E) Diagram depicting the modified neural specification protocol in which YSLC conditioned media is used to replace the SB431542+NOGGIN treatment in the conventional protocol for 11 days, before culturing hESCs in the presence of just FGF2 for 4 days. (F) Immunofluorescence of SOX2 (green), SOX1 (grey), and a merge of both in hESCs cultured in mTeSR (hESC) and hESCs cultured in YSLC conditioned media (+YSLC CM) for the first 11 days of the neural protocol before being placed in FGF2 as outlined in panel E (scale bar = 50 µm). (G) Bar charts (± SEM) depicting RFI of SOX1 and SOX2 in cells from panel F. Mann–Whitney U-test; *p<0.05, ****p<0.0001 (number of cells analysed: SOX1: hESC n = 4949, +YSLC CM n=4736, SOX2: hESC n = 4947, +YSLC CM n=4736, n=3 independent experiments). (H) Relative expression levels (REL) (± SEM) of *PAX6*, *NESTIN*, and *NOTCH1* in hESCs cultured as outlined in panel E normalised to hESC control. Mann-Whitney U-test ; *p < 0.05 (n=4 samples, two independent experiments).

The online version of this article includes the following source data and figure supplement(s) for figure 7:

**Source data 1.** Fluorescence intensity and qPCR analysis of anterior ectoderm and pluripotency markers in human embryonic stem cells (hESCs) in the presence of yolk sac-like cell (YSLC) conditioned medium (CM) in 2D and 3D.

**Figure supplement 1.** Yolk sac-like cells (YSLCs) induce the co-expression of anterior ectoderm markers and pluripotency markers in human embryonic stem cells (hESCs) in 2D and 3D.

**Figure supplement 1—source data 1.** Fluorescence intensity and qPCR analysis of anterior ectoderm and pluripotency markers in human embryonic stem cells (hESCs) in the presence of yolk sac-like cell (YSLC) conditioned medium (CM) in 2D and 3D.

We first sought to derive an *in vitro* stem cell line representative of the post-implantation human yolk sac. Following a screen of developmentally relevant signalling pathway activators, we identified concurrent ACTIVIN-A, WNT, and JAK-STAT pathway activation in hEPSCs, Rset hPSCs, and hESCs leads to an upregulation of pan-endodermal markers. This is in line with recent studies implicating these pathways in extra-embryonic endoderm specification (*Anderson et al., 2017*; *Linneberg-Agerholm et al., 2019*). Studies in human embryos have shown that inhibition of ACTIVIN-A signalling leads to a decrease in the number of epiblast and hypoblast cells (*Blakeley et al., 2015*), whereas studies in marmoset embryos demonstrated a role for the WNT pathway in hypoblast specification (*Boroviak et al., 2015*), indicating these signalling pathways are involved in hypoblast specification *in vivo*. Whether the JAK-STAT pathway is equally involved in hypoblast specification/maintenance in human and non-human primate embryos remains to be determined.

Our data shows that the pluripotent state of hPSCs determines their propensity to adopt an extra-embryonic endoderm- or definitive endoderm-like fate in response to ACL: Rset hPSCs, which harbour a pluripotent state that is an intermediate between naïve and primed (*Gafni et al., 2013*; *Kilens et al., 2018*), demonstrate extra-embryonic endoderm potential, whereas conventional hESCs, which are similar to the primed post-implantation epiblast (*Molè et al., 2021*; *Nakamura et al., 2016*), are more predisposed to definitive endoderm differentiation. This phenomena is also observed in mouse extra-embryonic endoderm specification; mouse ESCs are able to differentiate into extra-embryonic endoderm progenitors cells (XEN cells), whereas post-implantation primed EpiSCs are not (*Cho et al., 2012*). The observation that Rset hPSCs, which are transcriptionally similar to the implanting human epiblast, are still competent to form extra-embryonic endoderm is in line with the specification of human hypoblast progenitors at the time of implantation (*Niakan and Eggan, 2013*). Nonetheless, following ACL treatment, hEPSC, Rset hPSCs, and hESCs still maintained *OCT4* and *NANOG* expression, suggesting the cells still harbour some pluripotency features.

Interestingly, hEPSC treated with ACL give rise to a population of cells that harbour similarities to both the definitive endoderm and extra-embryonic endoderm. Overall hEPSC-ACL transcriptionally bear a greater similarity to definitive rather than extra-embryonic endoderm, and also harbour a large subpopulation of cells that undergo a mesendodermal intermediate state during endodermal specification, thereby resembling definitive endoderm specification. However, hEPSC-ACL are not as transcriptionally similar to the definitive endoderm as hESC-ACL are, and also demonstrate a propensity to contribute to the mouse primitive endoderm within interspecies chimeras. Collectively, this suggests that hEPSC-ACL may harbour a heterogenous population of endodermal cells that have varying degrees of similarity to either the definitive or extra-embryonic endoderm. This suggests that the starting hEPSC population may contain subpopulations with largely contrasting pluripotent states that span from naïve to primed. This is mirrored during human trophoblast stem cell induction wherein hEPSCs respond heterogeneously (*Castel et al., 2020*).

Pluripotent state also appeared to influence the developmental state of the extra-embryonic endoderm that hPSCs adopt in response to ACL treatment. Previously, T2iLGö-hPSCs, hPSCs similar to the pre-implantation human epiblast, have been shown to adopt an endodermal fate following treatment with ACTIVIN-A, the WNT activator Chiron, and LIF to give rise to 'T2iLGö-PrE' cells (*Linneberg-Agerholm et al., 2019*). By comparing pre- vs. post-implantation extra-embryonic endoderm gene expression signatures to Rset-ACL and T2iLGö-PrE, we found that T2iLGö-PrE were relatively more similar to the pre-implantation extra-embryonic endoderm, whereas Rset-ACL were more similar to the post-implantation extra-embryonic endoderm (yolk sac). T2iLGö hPSCs represent an earlier developmental pluripotent state than Rset hPSCs and may, therefore, be more predisposed to adopt an early extra-embryonic endoderm fate in response to ACL. However, Rset hPSCs represent a later, intermediate pluripotent state (*Kilens et al., 2018*) and may consequently adopt a more post-implantation extra-embryonic endoderm fate. Future investigations are needed to determine whether YSLCs can contribute to the yolk sac in human embryos.

To model the epiblast-yolk sac interactions *in vitro*, we used hESCs for the epiblast component of the model, as when cultured in 3D they are able to recapitulate key hallmarks of early post-implantation human embryos (*Moris et al., 2020*; *Shahbazi et al., 2017*; *Shao et al., 2017a*; *Simunovic et al., 2019*; *Taniguchi et al., 2015*; *Zheng et al., 2019*). We then determined YSLC's ability to influence cell fate in hESCs by analysing levels of anterior and posterior epiblast markers in hESCs in the presence of YSLCs or YSLC conditioned medium in both 2D and 3D contexts. This revealed that YSLCs inhibit WNT and BMP signalling in hESCs, thereby blocking posterior epiblast marker expression, while promoting the co-expression of pluripotency and anterior ectoderm markers.

Within mouse development, the AVE acts to antagonise signals from the trophoblast, leading to the establishment of the anterior-posterior axis across the epiblast (*Rivera-Pérez and Hadjantonakis, 2015*). Similarly, *in vitro* cultured human embryos harbour a subpopulation of anterior hypoblast cells that express BMP, WNT, and NODAL inhibitors and may, therefore, represent the human equivalent of an AVE signalling centre (*Molè et al., 2021*). YSLC's ability to influence hESC fate, promoting anterior ectoderm and pluripotency marker expression at the expense of posterior epiblast markers, via WNT and BMP pathway antagonism suggests a functionality similar to a mammalian AVE signalling centre.

Studies in the mouse embryo have shown that when the extra-embryonic ectoderm is removed such that only the epiblast and the visceral endoderm, including the AVE, remain, posterior epiblast fate is lost, with SOX1 and OCT4 being co-expressed within the epiblast (*Rodriguez et al., 2005*). Therefore, it is likely that this state is being mimicked when hESCs are cultured with YSLCs or in YSLC conditioned medium, as human trophoblast stem cells are missing from this model. The role that the human trophoblast plays in anterior-posterior patterning is not yet understood. Similarly, it is unclear whether primitive streak derivatives are required for complete anterior fate specification in the epiblast, as has been shown to be the case in the mouse (*Camus et al., 2000*). To assess whether the trophoblast is required for complete anterior ectoderm specification and, if so, whether it acts directly, via its secreted signals, or indirectly, via primitive streak specification, co-culture conditions supportive of the survival of YSLCs, hESCs, and human trophoblast stem cells need to be optimised. A model that incorporates the three cell types would be invaluable for dissecting anterior-posterior patterning in the human embryo.

Additionally, without more detailed transcriptional characterisation of the human AVE, it remains unclear whether YSLCs resemble human AVE specifically, or human post-implantation yolk sac generally. Given that in the primate post-implantation hypoblast, BMP and WNT inhibitors are initially expressed broadly before being restricted anteriorly (*Molè et al., 2021*; *Sasaki et al., 2016*), future in-depth transcriptional comparison and profiling of the peri-implantation human hypoblast should help to refine the positioning of YSLCs developmentally. Similarly, *LEFTY1* and *LEFTY2* are both upregulated in the human post-implantation hypoblast and in YSLCs, and so the role that NODAL signalling plays within anterior-posterior patterning remains to be explored.

In summary, our results implicate the human yolk sac in early human embryo morphogenesis and propose YSLCs as a useful tool for modelling the interaction between the post-implantation yolk sac and epiblast, providing a platform to dissect the mechanisms of human embryogenesis.

# Materials and methods

**Key resources table**

| Reagent type (species) or resource | Designation | Source or reference | Identifiers | Additional information |
|---|---|---|---|---|
| Cell line (*Homo sapiens*) | RUES2-GLR | Brivanlou Lab (Rockefeller Institute, New York City, NY) | | Human embryonic stem cell line |
| Cell line (*Homo sapiens*) | H9 | Vallier Lab (Stem Cell Institute, Cambridge, UK) under agreement with WiCell | (RRID:CVCL_1240) | Human embryonic stem cell line |
| Cell line (*Homo sapiens*) | SHEF6 | Shef-6 (UK Stem Cell Bank, NIBSC) | | Human embryonic stem cell line |
| Peptide, recombinant protein | Recombinant human ACTIVIN-A | Stem Cell Institute, Cambridge, UK | | 100 ng ml$^{-1}$ |
| Peptide, recombinant protein | Recombinant human LIF | Stem Cell Institute, Cambridge, UK | | 10 ng ml$^{-1}$ |
| Peptide, recombinant protein | Recombinant human WNT3a | R and D Systems | 5036-WN-010 | 1 µg ml$^{-1}$ |
| Peptide, recombinant protein | Recombinant human BMP4 | Peprotech | 120–05 | |
| Chemical compound, drug | CHIR99021 | Stem Cell Institute, Cambridge, UK | | 3 µM |
| Antibody | NANOG (goat polyclonal) | R and D Systems | AF1997 (RRID:AB_355097) | (1:200) |
| Antibody | SOX1 (goat polyclonal) | R and D Systems | AF3369-SP (RRID:AB_2239879) | (1:200) |
| Antibody | SOX2 (rabbit polyclonal) | Abcam | Ab59776 (RRID:AB_945584) | (1:200) |
| Antibody | T (goat polyclonal) | R and D Systems | AF2085 (RRID:AB_2200235) | (1:200) |
| Antibody | GATA6 (goat polyclonal) | R and D Systems | AF1700 (RRID:AB_2108901) | (1:200) |
| Antibody | LEF1 (rabbit monoclonal) | Cell Signaling Technology | 2230S (RRID:AB_823558) | (1:200) |
| Antibody | pSMAD1/5/9 (rabbit monoclonal) | Cell Signaling Technology | 13820S (RRID:AB_2493181) | (1:200) |
| Software, algorithm | ImageJ software | ImageJ (http://imagej.nih.gov/ij/) | RRID:SCR_003070 | |
| Software, algorithm | GraphPad Prism software | GraphPad Prism (https://graphpad.com) | RRID:SCR_015807 | |
| Other | 8-well µ-plates ibiTreat | Ibidi | 80826 | |
| Other | Rset (2-component) | StemCell Technologies | 05978 | |
| Other | mTeSR1 PLUS | StemCell Technologies | 05825 | |
| Other | Geltrex LDEV-free growth factor reduced | Thermo Fisher Scientific | A1413302 | |
| Other | Matrigel | BD Biosciences | 356230 | |

## Human embryonic stem cell culture

The UK Stem Cell Bank Steering Committee approved all hESC experiments. All experiments comply with the UK Code of Practice for the Use of Human Stem Cell Lines. The following hESC lines were used: RUES2-GLR (kindly provided by Ali Brivanlou, The Rockefeller University, New York City, NY), H9 (kindly provided by Ludovic Vallier, Stem Cell Institute, UK, under agreement with WiCell), and SHEF6 (kindly provided by the UK Stem Cell Bank, National Institute for Biological Standards and Control (NIBSC)). All hESC lines were maintained in a humidified incubator set at 37℃, 21% $O_2$, 5% $CO_2$. H9, SHEF6, and RUES2-GLR cell lines have been authenticated by STR profiling. All cell lines were tested for mycoplasma every 2 weeks and were all negative.

Conventional hESCs were either cultured on growth factor-reduced Matrigel-coated (Corning, 353046) dishes or on irradiated CF-1 mouse embryonic fibroblasts (MEFs) (ASF-1201, AMS

Biotechnology). For Matrigel coating, a 1.6% Matrigel solution in DMEM/F12 was incubated for 2 hr at room temperature (RT). When on Matrigel, hESCs were cultured in mTeSR1 (85850, StemCell Technologies), with medium changed every 24 hr. When cultured on MEFs, hESCs were cultured in 'Primed medium', consisting of DMEM F12 (21331–020, Thermo Fisher Scientific) supplemented with 100 µM β-mercaptoethanol (31350–010, Thermo Fisher Scientific), penicillin–streptomycin (15140122, Thermo Fisher Scientific), GlutaMAX (35050061, Thermo Fisher Scientific), MEM non-essential amino acids (11140035, Thermo Fisher Scientific), and 20% v/v KnockOut Serum Replacement (10828010, Thermo Fisher Scientific). This was supplemented with 12 ng ml$^{-1}$ bFGF2 (Stem Cell Institute) before use.

To convert hESCs to each of the states of pluripotency, cells were first plated from Matrigel onto MEFs in Primed Medium for 24 hr. To convert hESCs into 'extended pluripotent stem cells', hESCs were cultured in 'LCDM' medium as outlined in the initial publication (*Yang et al., 2017*) until domed colonies appeared. Briefly, LCDM consists of a basal N2B27 medium supplemented with 10 ng ml$^{-1}$ recombinant human LIF (Stem Cell Institute, Cambridge, UK), 1 µM CHIR 99021 (Stem Cell Institute, Cambridge, UK), 2 µM dimethindene maleate (1425, Tocris), 2 µM minocycline hydrochloride (sc-203339, Santa Cruz Biotechnology), 0.5 µM IWR-endo-1 (S7086, Selleckchem), and 2 µM Y27632 (72302, STEMCELL Technologies). N2B27 contains a 1:1 mix of DMEM F12 (21331–020, Thermo Fisher Scientific) and Neurobasal A (10888–022, Thermo Fisher Scientific) supplemented with 100 µM β-mercaptoethanol (31350–010, Thermo Fisher Scientific), penicillin–streptomycin (15140122, Thermo Fisher Scientific), GlutaMAX (35050061, Thermo Fisher Scientific), MEM non-essential amino acids (11140035, Thermo Fisher Scientific), 5% v/v KnockOut Serum Replacement (10828010, Thermo Fisher Scientific), 1% v/v B27 (10889–038, Thermo Fisher Scientific), and 0.5% of in-house produced N2, which consists of DMEM F12 medium (21331–020, Thermo Fisher Scientific), 2.5 mg ml$^{-1}$ insulin (I9287, Sigma-Aldrich), 10 mg ml$^{-1}$ Apo-transferrin (T1147, Sigma-Aldrich), 0.75% bovine albumin fraction V (15260037, Thermo Fisher Scientific), 20 µg ml$^{-1}$ progesterone (p8783, Sigma-Aldrich), 1.6 mg ml$^{-1}$ putrescine dihydrochloride (P5780, Sigma-Aldrich), and 6 µg ml$^{-1}$ sodium selenite (S5261, Sigma-Aldrich). Cells were passaged every 4–5 days, and medium was changed daily.

To convert to Rset hPSCs (05978, StemCell Technologies), hESCs were cultured in the commercially available medium as per the manufacturer's instructions. Cells were passaged once domed colonies had formed, and subsequently every 4–5 days, and medium was changed daily.

All cell lines were passaged using StemPro Accutase (A1110501, ThermoFisher Scientific), which was added for 3 min at 37°C, before being diluted in DMEM/F12 and centrifuged. Cells were then plated in their appropriate medium supplemented with 10 µM ROCK inhibitor Y-27632 (72304, STEMCELL Technologies). ROCK inhibitor was removed after 24 hr.

## Screen procedure and ACL conversion protocol

The initial chemical screen was carried out using RUES2-GLR hESCs. Inactivated MEFs were plated onto 96-well µ-plates ibiTreat (89626, Ibidi). The following day hEPSCs were plated in LCDM medium. After 24 hr, medium was replaced with LCDM medium supplemented with each of the variations of pathway activators listed in *Figure 1—figure supplement 1B*. Retinoic acid (R2625, Sigma-Aldrich) was used at varying concentrations (0.1, 1, 10 µM) along with the following concentrations for the other cytokines: 100 ng ml$^{-1}$ PDGF-AB (78096, StemCell Technologies), 20 ng ml$^{-1}$ human ACTIVIN-A (Stem Cell Institute, Cambridge, UK), 1 µM CHIR 99021 (Stem Cell Institute, Cambridge, UK), and 12 ng ml$^{-1}$ bFGF2 (Stem Cell Institute, Cambridge, UK). Medium was changed every 24 hr. After 6 days, each well was fixed in 4% paraformaldehyde (11586711, Electron Microscopy Sciences) at RT for 20 min and subsequently washed twice with 0.1% Tween-PBS in preparation for immunostaining. As a control, hEPSCs were cultured for 6 days in LCDM medium.

For the second screen, which sought to determine the effect of changing the basal medium, 8-well µ-plates ibiTreat were used (80826, Ibidi). hEPSCs were plated on MEFs in LCDM medium for 24 hr, before being replaced with each of the medium being screened for (conditions outlined in *Figure 1—figure supplement 1D*). Medium was changed every 24 hr and after 6 days cells were fixed as outlined above.

For proliferation experiments, hPSCs (Rset hPSCs, hESPCs, or hESCs) were plated on MEFs at density of 10,000 cells/well in a 24-well plate. Rset hPSCs, hEPSCs, and hESCs were plated in their respective medium. After 24 hr, this was replaced with either ACL medium + DMSO or ACi medium

(ACTIVIN-A + Chiron + 10 mM Jak inhibitor) (420097, Merck Millipore). After 24 hr of the addition of ACL+DMSO or ACi, the first proliferation timepoint was taken. This day was considered day 1 of hypoblast conversion. Cell number was counted using a haemocytometer every day during conversion with media change every day.

For hPSC ACL treatment, 250,000 of the respective cell lines were plated onto MEFs in 12 multi-well dishes (Corning, 353046) in their respective pluripotency medium. After 24 hr, medium was replaced with ACL. ACL consists of N2B27 basal medium supplemented with 100 ng ml$^{-1}$ human ACTIVIN-A (Stem Cell Institute, Cambridge, UK), 10 ng ml$^{-1}$ recombinant human LIF (Stem Cell Institute, Cambridge, UK), and 3 µM CHIR99021 (Stem Cell Institute, Cambridge, UK). Cells were passaged using Dispase (07923, StemCell Technologies) at a ratio of approximately 1:2–1:3 every 3–5 days.

## 3D cultures of hESCs and YSLCs

Geltrex LDEV-free growth factor reduced (A1413302, Thermo Fisher Scientific) and growth factor reduced Matrigel (356230, BD Biosciences) were thawed on ice and dispensed using ice-cold tips.

For co-cultures, 80 µl of Geltrex was added to 8-well µ-plates ibiTreat (80826, Ibidi) and incubated at 37°C for 4 min. Having been passaged using StemPro Accutase, 30,000 YSLCs and 30,000 hESCs were mixed in DMEM/F12 and plated on top of the Geltrex bed for 12 min at 37°C to allow the cells to attach to the matrix. Proceeding this, the DMEM/F12 was replaced with mTeSR1 supplemented with 5% Geltrex. Medium was changed every 24 hr with just mTeSR1. As a control 60,000 hESCs were plated in 3D Geltrex in mTeSR1, with medium changed daily. After 72 hr, the wells were fixed as outlined above.

## mTeSR conditioned medium experiments

To prepare YSLC conditioned medium , YSLCs were cultured until confluent in ACL. Medium was then changed to mTeSR1 PLUS (05825, StemCell Technologies) for 24 hr before being collected and filtered using syringe filters with a pore size of 0.45 µm (10109180, Thermo Fischer Scientific) and stored at 4°C for a maximum of a week. YSLC conditioned medium was used in two different sets of experiments:

- 3D culture: 80 µl of Geltrex was added to 8-well µ-plates ibiTreat (80826, Ibidi) and incubated at 37°C for 4 min; 30,000 hESCs resuspended in DMEM/F12 were then plated on top of the Geltrex bed for 12 min at 37°C. The DMEM/F12 was subsequently replaced with YSLC-conditioned medium supplemented with 5% Geltrex. Medium was changed every 24 hr with just YSLC conditioned medium. As a control 30,000 hESCs were plated in 3D Geltrex in mTeSR Plus, with medium being changed every 24 hr. After the determined length of culture, wells were fixed as outlined above.
- 2D culture: 30,000 hESCs were plated on Matrigel-coated 8-well µ-plates ibiTreat (as outlined in the section 'Human embryonic stem cell culture') in either YSLC conditioned medium, YSLC conditioned medium supplemented with 1 ng ml$^{-1}$ recombinant human BMP4 (120–05, Peprotech) or 1 µg ml$^{-1}$ WNT3a (5036-WN-010, R&D Systems), or MesoI YSLC conditioned medium. Medium was changed every 24 hr with supplemented agonist, and cells were fixed after 48 hr. Samples were fixed as outlined above.

Mesoderm induction conditioned medium experiments: hESCs were plated on Matrigel-coated 8-well µ-plates ibiTreat (see section 'Human embryonic stem cell culture') and cultured in mesoderm induction medium (MesoI) which was composed of N2B27 (as outlined above) supplemented with 10 ng ml$^{-1}$ of recombinant human BMP4 (120–05, Peprotech) for 3 days before being fixed. Alternatively, N2B27 was conditioned on confluent YSLCs for 24 hr before being supplemented with BMP4 and added to hESCs for 3 days before being fixed.

Neural differentiation conditioned medium experiments: hESCs were cultured in N2B27 that had been conditioned on confluent YSLCs for 24 hr on Matrigel-coated 8-well µ-plates ibiTreat (see section 'Human embryonic stem cell culture') for 11 days, before being passaged 1:2 and maintained in N2B27 supplemented with 20 ng ml$^{-1}$ bFGF2 (Stem Cell Institute, Cambridge, UK) for 4 days.

The neural differentiation protocol outlined in *Shi et al., 2012* was carried out as a positive control. Briefly, hESCs were cultured in N2B27 supplemented with 10 µM SB431542 (72232, Stem Cell Technologies) and 500 ng ml$^{-1}$ of NOGGIN (1967-NG, R&D Systems) on Matrigel-coated 8-well µ-plates ibiTreat (see section 'Human embryonic stem cell culture') for 11 days, before being passaged

1:2 and maintained in N2B27 supplemented with 20 ng ml$^{-1}$ bFGF2 (Stem Cell Institute, Cambridge, UK) for 4 days.

## Chimera experiments

All national and international guidelines were followed for mouse upkeep. Experiments were approved by the home office,reviewed by the University of Cambridge Animal Welfare and Ethical Review Body (AWERB), and were regulated by the Animals (Scientific Procedures) Act 1986 Amendment Regulations 2012. Animals were inspected daily and those that showed health concerns were culled by cervical dislocation. F1 (C57Bl6xCBA) females were superovulated by injection of 5 IU of pregnant mares' serum gonadotropin (PMSG, Intervet), followed by injection of 5 IU of human chorionic gonadotropin (hCG, Intervet) and mating with F1 males. Animals were culled by cervical dislocation.

For human cell-mouse embryo chimera experiments, eight-cell stage mouse embryos were recovered at embryonic day E2.5 prior to compaction by flushing the oviducts with M2. *Zona pellucida* was then removed by brief treatment with Acidic Tyrode's solution (T1788, Sigma-Aldrich). Concurrently, clumps consisting of three to six SOX17-tdTomato+ ACL-treated cells were manually picked. To create small clumps of cells, ACL-treated cells were dissociated in StemPro Accutase (07920, StemCell Technologies) for 1 min at 37°C, before being diluted in DMEM/F12 (11320033, Thermo Fischer Scientific), pipetted gently, centrifuged, and resuspended in ACL medium. The resulting clumps were then aggregated with the eight-cell stage mouse embryos in 100% ACL medium for 24 hr before being transferred into KSOM until E4.5. Embryos were then fixed as outlined above. For hEPSC and Rset hPSC controls, the same protocol was used, only ACL was replaced with the cell line's respective medium were relevant.

For all mouse embryo experiments, sample size was based on our own previous experience and relevant publications.

## Cell sorting

To isolate successfully converted hPSC-ACL cell populations for RNA-seq analysis, hESCs, Rset hPSCs, and hEPSCs were treated with ACL for 6 days. Cells were then dissociated using Accutase and filtered using syringe filters with a pore size of 0.45 μm before being resuspended in PBS supplemented with 5% fetal bovine serum (Stem Cell Institute, Cambridge, UK). The tdTomato+ population was then selected for by carrying out cell sorting using an AriaIII instrument, before being pelleted for RNA extraction.

## Library preparation/RNA-seq

The sequencing was performed on the Illumina HiSeq 4000 platform. Reads were aligned using the STAR algorithm (*Dobin et al., 2013*) and read counts of genes were quantified using the HTSeq algorithm (*Anders et al., 2015*) to calculate transcripts per million (TPM).

## RNA-seq analysis

Data from single-cell human hypoblast were obtained from *Zhou et al., 2019*, *Molè et al., 2021* and *Xiang et al., 2020*. Stem cell-derived definitive endoderm single-cell data was obtained from day 3 converted cells from *Cuomo et al., 2020*. Read one raw data from *Zhou et al., 2019* was trimmed using cutadapt and the adapter sequence was specified using '-g TGGTATCAACGCAGAGTACATGGG'. Raw data from Cuomo et al. were trimmed using TrimGalore! with default settings. Raw data from Xiang et al. was not trimmed. Reads were then aligned to the GRCh38 human reference transcriptome using kallisto in batch mode for smart-seq2 data (*Xiang et al., 2020*; *Cuomo et al., 2020*), and for *Zhou et al., 2019* kb-python was used with a custom whitelist for each set of fastq files and with custom technology specified as '-x 1,0,8:1,8,16,0,0,0'. Smart-seq2 transcript level counts were then collapsed into gene level counts as described elsewhere (*Booeshaghi et al., 2020*). Count tables were then filtered to include cells used in their respective publications. T2iLGö-PrE was obtained from *Linneberg-Agerholm et al., 2019* and processed to TPM. Data from (Molé et al., 2021) was processed an analysed as previously described.

Single-cell datasets were intersected based on common features. They were then integrated using Seurat v3 (*Stuart et al., 2019*) based on 10,000 variable features in 30 dimensions using

50,000 anchors to generate a merged Seurat object with batch-corrected counts in the 'integrated' assay. Cell identities were assigned based on single-cell metadata files provided by respective publications. The Seurat object was then subset based on identities of interest (epiblast vs. hypoblast vs. trophoblast vs. definitive endoderm; hypoblast vs. definitive endoderm-like cells, pre-implantation vs. post-implantation hypoblast). Markers to distinguish the identities were identified using the FindMarkers function on the subset objects, using a Wilcoxon ranked test on integrated counts with a log fold change minimum of 0.25 and a minimum of 10% of cells in a given identity expressing each positive marker. Genes with a p-value<0.05 were used for downstream analysis as transcriptome signatures. Using the average TPM of bulk RNA-seq cell-line data, GSVA was used to determine cell type similarity using marker gene sets (*Hänzelmann et al., 2013*). MA plots were created using ggplot2.

## RNA extraction

RNA was extracted with TRIzol reagent (15596010, Thermo Fisher Scientific). Briefly, snap-frozen cell pellets were suspended in 500 μL TRIzol, followed by a 5 min incubation at RT and addition of 0.1 mL chloroform. The samples were vortexed for 15 s, or until homogeneity in colour. After a 3 min incubation at RT, samples were centrifuged at 14,000 rpm for 15 min at 4°C to allow phase separation. The top aqueous RNA-containing layer was transferred to fresh tubes; 1 μL RNAse-free glycogen (AM9510, Thermo Fisher Scientific) and 250 μL isopropanol (I91516, Sigma) were added. Samples were mixed by inverting three to four times and incubated for 10 min at RT. A 30 min centrifugation at 14,000 rpm at 4°C was performed to pellet the RNA, which was subsequently washed with 250 μL of 75% ethanol. The supernatant was removed, and the pellet air-dried before suspending in 50 μL nuclease-free water. The samples were vortexed for 15 s and then incubated at 55°C for 10 min to completely dissolve the RNA. The samples were stored at −80°C.

## Retrotranscription and RT-PCR analysis

To make sample cDNA, 1 μg of RNA was used to perform a reverse transcriptase reaction in the presence of random primers (C1181, Promega), dNTPs (N0447S, New England BioLabs), RNase inhibitor (M0314L, New England BioLabs), and M-MuLV reverse transcriptase (M0253L, New England BioLabs).

RT-PCR reactions were carried out using Power SYBR Green PCR Master Mix (4368708, Thermo Fisher Scientific) on a Step One Plus Real-Time PCR machine (Applied Biosystems). The following programme was used: 10 min at 95°C and 40 cycles of 15 s at 95°C and 1 min at 60°C. qPCR data was normalised against expression levels of hypoxanthine guanine phosphoribosyltransferase (*HPRT*), which should be consistent across samples. Error bars represent variability (SEM) across multiple independent experiments. A complete list of primers used is shown in Table S4.

## Immunostaining and imaging

Permeabilisation was performed using PBS with 0.1 M glycine, 0.3% Triton X-100 for 30 min at RT. Primary antibodies were incubated overnight at 4°C. All primary antibodies were diluted in blocking buffer (1% BSA, 0.1% Tween in PBS) at 1:200. Cells were then washed three times with 0.1% Tween-PBS before being incubated with fluorescently conjugated Alexa Fluor secondary antibodies diluted 1:500 in 0.1% Tween-PBS for 2 hr at room temperature. Where indicated Alexa Fluor 647 Phalloidin (1:100; A22287, Thermo Fisher Scientific) and DAPI (1:1000; D3571 Thermo Fisher Scientific) were added concomitantly with the secondary antibodies. Primary and secondary antibody lists can be found in *Supplementary file 4*.

Imaging took place on an inverted SP5 confocal microscope (Leica Microsystems) using a Leica HC PL FLUOTAR 0.5 NA 20.0× dry objective and a Leica HC PL APO 1.4 NA 63× oil objective, or on an inverted SP8 confocal microscope (Leica Microsystems) with a Leica HC PL APO CS2 1.4 NA 63× oil objective and a Leica Fluotar VISIR 0.95 NA 25× water objective. Within each independent experiment, laser power and detector gain were maintained constant.

## Image analysis

All microscopy analysis was carried out using FIJI software (http://fiji.sc) (*Schindelin et al., 2012*). For all quantifications, laser power and detector gain were maintained constant to quantitatively compare different experimental conditions within a single experiment. A single representative Z-plane was chosen for all quantifications. When calculating protein levels within a cell population, 10 microscopic fields of view were randomly captured for each independent experiment to obtain representative coverage of the cell population within each experiment.

### Quantification of nuclear fluorescence intensity

Nuclear shape was conferred from DAPI staining, images of which were binarised to produce a mask to segment the nuclei in 2D. The watershed function on FIJI was then used to separate nuclei that were closely positioned. Each nuclear outline was then saved as a region of interest (ROI) and the fluorescence intensity within each ROI was then analysed. Within each experiment, the experimental fluorescence intensity values were normalised to the control expression level for each protein being analysed.

### Classification of spheroid morphology

Within co-culture and conditioned medium experiments, each structure was classified as disorganised, squamous epithelial or columnar epithelial. 'Disorganised structures' were those which contain no discernible epithelial structures and were mostly a 2D 'carpet' of cells. Squamous epithelial spheroids were mainly composed of cells with an aspect ratio (height/width) of <1. Columnar epithelial spheroids were mainly composed of cells with an aspect ratio >1.

## Statistical analyses

All statistical analysis other than that involved in sequencing data analysis was carried out using GraphPad Prism software. Qualitative data are presented as a contingency table and were analysed with a $\chi^2$ test. Quantitative data are presented as mean ± SEM. Data with a Gaussian distribution were analysed using a two-tailed unpaired Student's t-test when comparing two groups or ANOVA with a Tukey's or Sidak's multiple comparison test when comparing multiple. If there were significant differences in the variance, a Welch's correction was used. If the data did not have a Gaussian distribution, a Mann–Whitney U-test was used when comparing two groups or a Kruskal–Wallis test with a Dunn's multiple comparison test was used when comparing multiple groups. Independent experiments are defined as biological replicates. Technical replicates were undertaken simultaneously.

## Acknowledgements

The work in the MZG laboratory is supported by grants from the Wellcome Trust (207415/Z/17/Z), the ERC advanced grant (669198), the NIH R01 (HD100456-01A1) grant, the NIH Pioneer Award (DP1 HD104575-01), Open Philanthropy/Silicon Valley Community Foundation, Weston Havens Foundation, and Shurl and Kay Curci Foundation to MZG. KM is a recipient of PhD studentship funding from the BBSRC. BATW is a recipient of a PhD studentship funded by the Gates Cambridge Trust. CEH is the recipient of a PhD studentship funded by The Centre for Trophoblast Research (University of Cambridge). Work in the group of MNS is funded by the Medical Research Council (MRC, award number MC_UP_1201/24) and the European Molecular Biology Organisation (Advanced EMBO fellowship). This research was also supported by the Cambridge NIHR BRC Cell Phenotyping Hub, who we thank for their advice and support in flow cytometry/cell sorting/imaging.

## Additional information

### Funding

| Funder | Grant reference number | Author |
|---|---|---|
| Biotechnology and Biological Sciences Research Council | 1943755 | Kirsty ML Mackinlay |
| Medical Research Council | MC_UP_1201/24 | Marta N Shahbazi |

| European Molecular Biology Laboratory | | Marta N Shahbazi |
|---|---|---|
| Wellcome Trust | 207415/Z/17/Z | Magdalena Zernicka-Goetz |
| National Institutes of Health | HD100456-01A1 | Magdalena Zernicka-Goetz |
| European Research Council | 669198 | Magdalena Zernicka-Goetz |
| National Institutes of Health | DP1 HD104575-01 | Magdalena Zernicka-Goetz |
| University of Cambridge | | Charlotte E Handford |
| Gates Cambridge Trust | | Bailey AT Weatherbee |
| European Molecular Biology Organization | | Marta N Shahbazi |

The funders had no role in study design, data collection and interpretation, or the decision to submit the work for publication.

## Author contributions

Kirsty ML Mackinlay, Conceptualization, Data curation, Formal analysis, Investigation, Visualization, Methodology, Writing - original draft, Project administration, Writing - review and editing; Bailey AT Weatherbee, Formal analysis, Investigation, Visualization, writing- review and editing; Viviane Souza Rosa, Data curation, Formal analysis, Validation, Investigation; Charlotte E Handford, Investigation, Data curation; George Hudson, Investigation; Tim Coorens, Formal analysis; Lygia V Pereira, Sam Behjati, Supervision; Ludovic Vallier, Resources, Supervision; Marta N Shahbazi, Conceptualization, Resources, Supervision, Investigation, Methodology, Project administration, Writing - review and editing; Magdalena Zernicka-Goetz, Supervision, Writing - review and editing

## Author ORCIDs

Kirsty ML Mackinlay https://orcid.org/0000-0001-6276-4274
Bailey AT Weatherbee https://orcid.org/0000-0002-6825-6278
George Hudson http://orcid.org/0000-0003-2506-6965
Ludovic Vallier http://orcid.org/0000-0002-3848-2602
Marta N Shahbazi https://orcid.org/0000-0002-1599-5747
Magdalena Zernicka-Goetz https://orcid.org/0000-0002-7004-2471

## Ethics

hESC experiments: The UK Stem Cell Bank Steering Committee approved all hESC experiments. All experiments comply with the UK code of Practise for the Use of Human Stem Cell Lines. Mouse embryo work: All national and international guidelines were followed for mouse upkeep. Experiments were approved by the Home Office, reviewed by the University of Cambridge Animal Welfare and Ethical Review Body (AWERB), and were regulated by the Animals (Scientific Procedures) Act 1986 Amendment Regulations 2012. Animals were inspected daily and those that showed health concerns were culled by cervical dislocation.

## Decision letter and Author response

Decision letter https://doi.org/10.7554/eLife.63930.sa1
Author response https://doi.org/10.7554/eLife.63930.sa2

# Additional files

## Supplementary files

• Supplementary file 1. Transcriptional signature of human embryonic and extra-embryonic lineages.

• Supplementary file 2. Transcriptional signature of human extra-embryonic endoderm and definitive endoderm.

• Supplementary file 3. Transcriptional signature of human pre- and post-extra-embryonic endoderm.

- Supplementary file 4. Supplementary materials and methods.
- Transparent reporting form

## Data availability

Raw data is available on EGA at accession number: EGAD00001006056 under the study number EGAS00001003571.

The following dataset was generated:

| Author(s) | Year | Dataset title | Dataset URL | Database and Identifier |
|---|---|---|---|---|
| Mackinlay K, Rosa VS, Weatherbee B, Hudson G, Coorens TT, Pereira LV, Behjati S, Vallier LL, Shahbazi M, Zernicka-Goetz M | 2020 | Understanding Self-Organising Capacity of Stem Cells during Implantation and Early Post-implantation Development in vitro and in vivo: Implications for Human Development (2020-04-20) | https://ega-archive.org/datasets/EGAD00001006056?fbclid=IwAR3-G8UAMWC-2XCkvM9rVh6H1NhT-w40PX4UF78E0yP4YmJ-Qy4l8sEhCDLtn8 | EGA,European Genome-phenome Archive, EGAD00001006056 |

The following previously published datasets were used:

| Author(s) | Year | Dataset title | Dataset URL | Database and Identifier |
|---|---|---|---|---|
| Zhou F, Wang R, Yuan P, Ren Y, Mao Y, Li R, Tang F | 2019 | Reconstituting the transcriptome and DNA methylome landscapes of human implantation | https://www.ncbi.nlm.nih.gov/geo/query/acc.cgi?acc=GSE109555 | NCBI Gene Expression Omnibus, GSE109555 |
| Xiang L, Yin Y, Zheng Y, Ma Y, Li Y, Zhao Z, Li T | 2019 | A developmental landscape of 3D-cultured human pre-gastrulation embryos | https://www.ncbi.nlm.nih.gov/geo/query/acc.cgi?acc=GSE136447 | NCBI Gene Expression Omnibus, GSE136447 |
| Cuomo ASE, Seaton DD, McCarthy DJ, Martinez I, Bonder MJ, Garcia-Bernardo J, Consortium H | 2020 | Single_Cell_RNAseq_at_various_stages_of_HiPSCs_differentiating_toward_definitive_endoderm_and_endoderm_derived_lineages | https://www.ebi.ac.uk/ena/browser/view/PRJEB14362 | ENA, ERP016000 |
| Linneberg-Agerholm M, Wong YF, Herrera JAR, Monteiro RGV, Anderson K, MBrickman J | 2019 | Naïve human pluripotency stem cells respond to Wnt, Nodal and LIF signalling to produce expandable naïve extra-embryonic endoderm | https://www.ncbi.nlm.nih.gov/geo/query/acc.cgi?acc=GSE138012 | NCBI Gene Expression Omnibus, GSE138012 |
| Yang Y, Liu B, Xu J, Wang J, Wu J, Shi, C, Deng H | 2017 | Global transcriptional analysis and genome-wide analysis of chromatin state in extended pluripotent stem cells, primed pluripotent stem cells, and naïve pluripotent stem cells | https://www.ncbi.nlm.nih.gov/geo/query/acc.cgi?acc=GSE89303 | NCBI Gene Expression Omnibus, GSE89303 |
| Molè MA, Coorens THH, Shahbazi MN, Weberling A, Weatherbee BAT, Gantner CW, Sancho-Serra C, Richardson L, Drinkwater A, Syed N, Engley S, Snell P, Christie L, Elder K, Campbell A, Fishel S, Behjati S, Vento-Tormo R, Zernicka-Goetz M | 2021 | Single cell RNA-seq of in vitro cultured human embryos | https://www.ebi.ac.uk/arrayexpress/experiments/E-MTAB-8060/ | ArrayExpress, E-MTAB-8060 |

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
