## [Decision Letter]

**Acceptance summary:**

In this work, Mackinlay et al. established a protocol to differentiate from human embryonic stem cells (ESCs) an extraembryonic cell type that resembles post-implantation hypoblast. They further demonstrate that these yolk sac-like cells induce in human ESCs expression of pluripotency and anterior ectoderm markers while repressing mesoderm and endoderm markers, by releasing BMP and WNT signaling pathway inhibitors. The human yolk sac-like cells generated here provide a new tool for in vitro investigations of human embryogenesis.

**Decision letter after peer review:**

Thank you for submitting your article "An in vitro stem cell model of human epiblast and yolk sac interaction" for consideration by *eLife*. Your article has been reviewed by 2 peer reviewers, and the evaluation has been overseen by a Reviewing Editor and Kathryn Cheah as the Senior Editor. The following individual involved in review of your submission has agreed to reveal their identity: Jacob H. Hanna (Reviewer #1).

The reviewers have discussed the reviews with one another and the Reviewing Editor has drafted this decision to help you prepare a revised submission.

Summary:

In this work, Mackinlay et al., developed an in vitro model for human early embryogenesis by establishing a protocol to differentiate from hESCs an extraembryonic cell type that resembles post-implantation yolk sac cells. They subsequently employed conditioned media from these yolk sac-like cells in combination with hESCs to analyze signaling interactions between humans ESCs and hypoblast cells. The authors present evidence to support the induction of extraembryonic endoderm features, which are dependent of the initial cell pluripotency state. Based on transcriptomic data, they argue that the hypoblast cells derived using the Rset media resemble those cells found in the human post-implantation hypoblast. Finally, the authors employed these human hypoblast-like cells to dissect the role of Wnt and BMP signaling pathways in yolk sac and hESC cells co-cultures, and identify the induction of an anterior ectoderm identity in hESCs by the yolk sac cells. The manuscript provides an advance in generation of extra-embryonic endoderm from hPSCs and in investigating in vitro interactions between embryonic and extraembryonic tissues during early human development.

Essential revisions:

There was consensus among the reviewers and editor that this model of human yolk sac cells represents a highly valuable system for examining the human hypoblast, especially given the difficulties to obtain post-implantation human embryos. In addition, the hPSCs co-culture protocol established herein provides novel functional evidence of an AVE-like role of human hypoblast. However, the reviewers also expressed several concerns about the data and their presentation and agreed that the current data do not fully support the authors' conclusions.

In particular, the key conclusion on line 358-359 "We will refer to Rset-ACL cells as to yolk sac cells" as well as other such conclusions in the manuscript are not fully supported by the data in the current manuscript. Either additional experimental analyses are needed, or this conclusion needs to be toned down throughout the manuscript. Referring to these cells as "yolk sac-like cells" would be more aligned with the current data and appropriate. In the revised manuscript, the authors are encouraged to highlight the caveats of their model and state that further improvements are needed with details.

– Identified markers that distinguish DE and PE using published data comparing data from wildly different sources. How were the different sources dealt with? What specific statistical cut-offs? The authors conclude that one cell culture product is more like PE than the other two. Does this make it PE? Overall correlation pretty weak (0.2). Similarly, in chimera experiments, the contribution was observed only 15% of the time, and only 1 or 2 cells at a time. Does this make it PE? As stated above, these concerns would need to be addressed or the relevant conclusions on the derivation of yolk sac cells should be toned down.

– Evidence of AVE-like character is too indirect: conditioned medium prevents BMP-directed hESC differentiation as measured by *SOX2* alone (single marker, not a robust assay) and pSMAD and same for WNT-directed hESC differentiation (and reduced LEF1 levels). The comparison to the AVE also warrants more discussion, as the murine AVE represents a subset of extra-embryonic endoderm, what explains asymmetry establishment in the epiblast. Hence, the yolk sac-like cells the authors describe are unlikely to represent the entire hypoblast of post-implantation human embryo, or how such yolk sac cells could break asymmetry in the epiblast is not clear.

– Figure 1 panel K: Nanog expression levels are increase after ACL treatment, when Nanog is generally considered a marker of the epiblast. Could the authors explain this discrepancy?

– Figure 2 panel D: Rset-ACL cells have a gene expression signature more similar to hypoblast, while hEPSC-ACL seem to be similar to both extraembryonic and definitive endoderm. Nevertheless, as mentioned before (line 164-165), hEPSCs-ACL are segregated into different subpopulations expressing different levels of Gata6/*Sox2*. Did the authors consider these differences in the population when performing bulk RNA-seq? Is it possible that eliminating the Gata6low/Sox2high subpopulation improves the induction to extraembryonic endoderm from EPSCs?

– Figure 3 panel A-B: In the quantification of Brachyury+ cells during ACL treatment (panel A), in the case of hESCs at day 2 there is about 30% of Brachyury+ cells, however in the immunofluorescence show in panel B for hESCs at day 2 the percentage of Brachyury+ cells seems to be much higher than 30%. Could the authors explain this discrepancy? Maybe a quantification of the relative fluorescence intensity as done in Figure 1 will be helpful to clarify this point, since it might be difficult to discern between positive and negative cells as there is a gradient of signal intensity.

–Figure 6 panel E and H. The authors showed that culturing hESCs with yolk sac cell-conditioned media in 3D Geltrex settings prevents *Sox2* downregulation and promotes a columnar spheroid morphology (line 387 in the text). In figure 6, they tested whether secretion of BMP and Wnt antagonist by yolk sac cells are responsible for preventing pluripotency exit in 3D (line 415 in the text), by treating hESCs with BMP4/Wnt3a and yolk sac-conditioned media in 2D cultures, nevertheless, the author do not comment about what is the effect of these conditions in the 3D context, given that the formation of columnar epithelial spheroids is also promoted with yolk sac-conditioned media. Do the cells form predominantly columnar/squamous spheroids or do they lose their integrity? Also, *Sox2* is a known marker for ectodermal differentiation, did the authors test what is the effect of adding BMP4/Wnt3a and yolk sac-conditioned media in other pluripotency markers like Oct4 or Nanog?

– Figure 7. The dynamics of ectodermal and pluripotency markers seem insufficient to support the idea of stable induction of anterior ectoderm. Sox1 is high since day 1 and Otx2 goes down at day 5. Is it possible that these markers get stabilized after more than 5 culture days? The authors should analyze the expression of mesodermal and endodermal markers to show that these markers are not induced.

Revisions expected in follow-up work:

Additional experimental data in support of the identify of Rec-ACL cells as genuine post implantation yolk sac cells, and discerning their activity as equivalent to the murine AVE.

[Editors' note: further revisions were suggested prior to acceptance, as described below.]

Thank you for resubmitting your work entitled "An in vitro stem cell model of human epiblast and yolk sac interaction" for further consideration by *eLife*. We apologize that the review process took longer than we strive for at *eLife*.

Your article has been reviewed by 2 peer reviewers, and the evaluation has been overseen by a Reviewing Editor and Kathryn Cheah as the Senior Editor. The following individual involved in review of your submission has agreed to reveal their identity: Jacob H. Hanna (Reviewer #1).

The manuscript has been improved and the authors toned down their conclusions about generating Yolk Sac Cells referring to them Yolk Sac Like Cells (YSLC). However, there are some remaining significant issues that need to be addressed, as outlined below:

1. The revised manuscript did not fully convince the reviewers that the ACL protocol represents a novel way to produce human extraembryonic endoderm.

Lines 192-194: "This revealed that all three cell populations bore a greater similarity to the hypoblast following ACL^-^treatment, while their similarity to the epiblast decreased, suggesting a shift towards an endodermal fate (Figure 1L)."

One is concerned the data simply show that sorting SOX17-positive cells from a mixed population of differentiating stem cells can enrich the endodermal transcriptional signature. Why is it surprising that the SOX17-positive subpopulation of differentiated stem cell lines bears similarity to endoderm, when SOX17 is known to be both necessary and sufficient for endodermal differentiation? In other words, does the novel ACL treatment protocol enhance the endodermal qualities of SOX17-positive cells that can be produced from stem cell lines by any other differentiation protocol? Given that the goal here is to establish a protocol for generating hypoblast-like cells from hESCs and the current methods direct hESCs towards definitive endoderm fate, it would be important to include in Figure 1L also definitive endoderm for comparison.

2. The conclusion that pluripotent state dictates response to the ACL treatment is underdeveloped, owing to flaws in experimental design.

In fact, the authors observe major differences between biological replicate hEPSC lines in terms of their response to ACL treatment (compare Figure 1D, top Gata6, Sox17 and Figure 1 Supp 2 K, top Gata6 and Sox17. Additionally, Figure 1 Supp 2 D shows very poor resolution of *Sox2* and Gata6 expressing populations after ACL treatment of this cell line). To their credit, the authors note on Line 173: "This suggests that there is a heterogeneous response to ACL^-^treatment within hEPSCs." Yet they then drop this cell line and exclude it from subsequent analyses.

Therefore, how can the authors be confident that the reported transcriptional differences captured in Figure 1L (among ACL^-^treated stem cell lines) is meaningful when they haven't really captured variation among biological replicate hEPSC cell lines?

In other words, the data shown in Figure 1L are cherry-picked because they exclude the cell line that did not perform as desired (and shown in Supp. 2).

3. The evidence that the authors have identified a cell type that, on the basis of transcriptome analysis, is a good model for extraembryonic endoderm is also cherry-picked because the authors focus on merely 40 genes to evaluate transcriptional phenotypes. Admittedly, selection of these 40 genes was conducted by rationale that is logical and justified. But the subsequent use of these 40 genes to compare treatments merely ranks the cell lines to each other. The analysis does not produce insight into whether these treatments are wholistically producing extraembryonic endoderm.

Moreover, even if we take the authors' word, Line 244 states:

"Rset-ACL the most relatively similar to extra-embryonic endoderm (Figure. 2C)." Similarly, Line 245 states, "hEPSC-ACL were more similar to extra-embryonic endoderm than to definitive endoderm."

Yet examination of Figure 2B does not strongly support these statements. For example, from 20 extra-embryonic endoderm genes only 6 appear to be upregulated, and from 20 definitive endoderm genes 2 are downregulated for Rset-ACL. As the differences in the number of upregulated and down regulated extra embryonic or definitive endoderm genes do not clearly identify a cell type fully aligned with hypoblast, this raises further concerns that the ACL treated cell populations are likely heterogenous rather than representing one YCLC cell type.

4. Chimera experiments are lacking controls.

What is the degree chimerism when Rset and hEPSC are not cultured in ACL? Given the very minor degree chimerism (3D and 3G) and the evidence of incomplete population-level differentiation of Rset and hEPSC in ACL (see above), the possibility remains that the authors are simply testing the already-proved hypothesis that Rset and hEPSC can chimerise extraembryonic endoderm in mouse embryos (Gafni et al., 2013; Yang et al., 2017). Moreover, in Figure 3, the number of chimera embryos analyzed nor the number of experiments are not provided.

5. Which cell line is the most similar to postimplantation hypoblast? Still unclear.

A transcriptional comparison was performed in Figure 4. However, the same weaknesses detailed in Point 3 above also apply here. In short, the transcriptional differences shown do not compellingly support the authors' conclusions.

A functional analysis was performed in Figure 4 Supp. 2 in postimplantation chimeras. However, only a single cell line was evaluated (only the cell line they claim to be the most postimplantation). Therefore, this experiment lacks appropriate controls. Additionally, it is unclear why the authors chose to mimic mouse postimplantation ex vivo, when the standard is to transfer the chimeras to recipient mice to observe actual postimplantation stages.

6. Evaluation of "cell fate" in Figure 5 and 6 is superficial.

Only a single pluripotency marker (*SOX2*) and a single mesendoderm marker (T) are analysed. Therefore, the results are still preliminary and conclusions weaker than they should be.

In these experiments, gene expression is analyzed primarily at the protein level by relative fluorescence intensity (RFI). As DAPI (or other nuclear marker) images are not shown, is RFI relative to DNA label? This should be clarified.

7. It is not clear how stable over time are YSLC when cultured in ACL?

The Discussion should more clearly articulate the limitations of the YSLC model and that hEPSCs gave rise to a population of cells that bears a similarity to the extra-embryonic endoderm but may also contain a definitive endoderm- like subpopulation.

---

## [Author Response]

Essential revisions:There was consensus among the reviewers and editor that this model of human yolk sac cells represents a highly valuable system for examining the human hypoblast, especially given the difficulties to obtain post-implantation human embryos. In addition, the hPSCs co-culture protocol established herein provides novel functional evidence of an AVE-like role of human hypoblast. However, the reviewers also expressed several concerns about the data and their presentation and agreed that the current data do not fully support the authors' conclusions.In particular, the key conclusion on line 358-359 "We will refer to Rset-ACL cells as to yolk sac cells" as well as other such conclusions in the manuscript are not fully supported by the data in the current manuscript. Either additional experimental analyses are needed, or this conclusion needs to be toned down throughout the manuscript. Referring to these cells as "yolk sac-like cells" would be more aligned with the current data and appropriate. In the revised manuscript, the authors are encouraged to highlight the caveats of their model and state that further improvements are needed with details.

We thank the reviewers for their appreciation of the value of our work and for their insightful comments, which have allowed us to greatly improve our manuscript. We agree that referring to our cells as “Yolk-sac like cells” (YSLCs) is more appropriate and so we have adjusted this throughout the manuscript. The reviewers’ comments provided us with an opportunity to further characterise the interaction between YSLCs and hESCs and as such further analyse the functional similarities between YSLCs and the AVE has been carried out. In our revised manuscript we provide:

– Evidence that YSLCs can contribute to the extra-embryonic endoderm compartment post-implantation mouse/human interspecies chimeras (new Figure 4—figure supplement 2).

– Analysis of the levels of LEF1 and pSMAD1/5 in hESCs cultured in 3D in the presence of YSLC conditioned medium, revealing that YSLC conditioned medium is able to inhibit both WNT and BMP signalling activation in hESCs, respectively (new Figure 5F-H).

– Analysis of the levels of NANOG in hESCs cultured in the presence of either BMP4, and BMP4+YSLC conditioned medium, or WNT3A, and WNT3A+YSLC conditioned medium, revealing that YSLC conditioned medium is able to inhibit the effect of both signalling pathway activators (new Figure 5—figure supplement 1H-K).

– Analysis of the ability of YSLCs to inhibit endoderm and mesoderm specification in hESCs in both a 2D and 3D context, revealing that YSLCs are able to inhibit the upregulation of both mesoderm and endoderm markers in hESCs (new Figure 6).

– Evidence that YSLCs are able to induce the upregulation of anterior ectoderm markers in hESCs in 2D (new Figure 7F-H).

– In addition, in the revised discussion we acknowledge the limitations of our model and which future improvements will be needed.

– Identified markers that distinguish DE and PE using published data comparing data from wildly different sources. How were the different sources dealt with? What specific statistical cut-offs?

We thank the reviewers from bringing this to our attention. In the original version of our manuscript we did not realign the datasets to a common reference genome. Instead, datasets were integrated using Seurat (Stuart et al., 2019). Gene abundances were then converted to relative abundances per million (RPCM) for downstream analysis and visualization. However, we have now taken raw data from the three different datasets (Cuomo et al., 2020; Xiang et al., 2019; Zhou et al., 2019), removing Stirparo et al. (2018), and realigned each of them using kallisto/kb-python to the GRCh38 reference transcriptome. The UMI counts/estimated counts were then integrated using Seurat v3 based on 10,000 variable features in 30 dimensions using 50,000 anchors to generate a merged Seurat object with batch corrected counts in the "integrated" assay.

As a result of this reanalysis, the differentially expressed genes identified for definitive endoderm, extra-embryonic endoderm, primitive endoderm and post-implantation hypoblast have changed (new Figure 2 A, B and Figure 4 A, B). This has also adjusted the values for the Z-score-based transcriptional analysis (new Figure 2C and Figure 4C) but has not impacted the overall conclusions drawn from this analysis. The process of realignment has strengthened the reliability of our analysis as it made the datasets more comparable.

The authors conclude that one cell culture product is more like PE than the other two. Does this make it PE? Overall correlation pretty weak (0.2).

Our Z-score-based transcriptional profiling involves placing the analysed samples within a relative scale of similarity, whereby definitive endoderm exists at one end, and extra-embryonic endoderm exists at the other. This analysis allows us to determine the relative similarity across the three cell types being analysed, telling us which, relative to the other cell types rather than in absolute terms, was the most transcriptionally similar to the extra-embryonic endoderm of the three. A positive value indicates a relative similarity to extra-embryonic endoderm, while a negative value suggests a relative similarity to definitive endoderm. For simplicity, we have normalised the scale to -1 and 1. The new values of similarity for Rset-ACL and hEPSC-ACL are 0.48 and 0.41, respectively (Figure 2C). While hESC-ACL had a similarity value of -0.89 (Figure 2C). Given that the analysis is relative, Rset-ACL is the most extra-embryonic endoderm-like relative to the other two cells types, whereas hESC-ACL is the most similar to definitive endoderm of the three cell types. This is also the case when using Z-score-based transcriptional analysis for determining similarity to primitive endoderm vs post-implantation hypoblast. For this reason, we agree with the reviewers’ that we cannot claim we have generated yolk sac cells in absolute terms. Instead, we now refer to Rset-ACL cells as yolk sac-like cells.

Similarly, in chimera experiments, the contribution was observed only 15% of the time, and only 1 or 2 cells at a time. Does this make it PE? As stated above, these concerns would need to be addressed or the relevant conclusions on the derivation of yolk sac cells should be toned down.

Interspecies chimeras yield relatively low contribution relative to same-species chimeras. Previous mouse-human chimera attempts using human pluripotent stem cells that do not involve inhibiting the apoptotic pathway have yielded blastocyst contribution between 0-15% (Huang et al., 2018; James et al., 2006; Masaki et al., 2015; Yang et al., 2017). Therefore, the level of contribution of hEPSC-ACL and Rset-ACL is comparable to previously observed levels of human-mouse chimeras. We agree that contribution to the primitive endoderm compartment alone is not evidence of an extraembryonic fate. However, it suggests that the cell types possess characteristics that are similar to extra-embryonic endoderm in order for them to survive and incorporate to the primitive endoderm compartment within the embryo.

Given that Rset-ACL were the most similar to the post-implantation hypoblast, we cultured the chimeric embryos to post-implantation stages in vitro. This revealed that 3 days after chimeric blastocysts were transferred to a post-implantation culture method (Ma et al., 2019), 9% of all viable embryos contained HuNu+/GATA6+ cells (new Figure 4—figure supplement 2A, B), suggesting that YSLCs are able to integrate into the post-implantation extra-embryonic endoderm compartment of the mouse embryo, further demonstrating that YSLCs harbour a yolk sac-like phenotype. These experiment were carried out, stained and imaged with the support of Charlotte E. Handford and so her name has been added to the author list. The other authors have agreed to this addition.

– Evidence of AVE-like character is too indirect: conditioned medium prevents BMP-directed hESC differentiation as measured by Sox2 alone (single marker, not a robust assay) and pSMAD and same for WNT-directed hESC differentiation (and reduced LEF1 levels). The comparison to the AVE also warrants more discussion, as the murine AVE represents a subset of extra-embryonic endoderm, what explains asymmetry establishment in the epiblast. Hence, the yolk sac-like cells the authors describe are unlikely to represent the entire hypoblast of post-implantation human embryo, or how such yolk sac cells could break asymmetry in the epiblast is not clear.

We agree with the reviewers’ comments that analysing SOX2 alone in response to YSLC conditioned medium is not sufficient. Analysing SOX2 dynamics in response to YSLCs acted as an initial indicator of whether YSLCs could influence hESC specification and, if so, whether this was orchestrated by WNT and BMP inhibition. We have adjusted Figure 5 so that this is clearer by first addressing the levels of SOX2 within hESCs in response to YSLCs in 3D, before analysing the role of WNT and BMP in mediating this response. This involved combining data from the old Figure 5 and old Figure 6. We have also added a new experimental data (new Figure 6 and new Figure 7) in which we analyse the levels of endoderm, mesoderm and ectoderm markers in hESCs cultured in 2D in the presence of YSLC conditioned medium as described below in more detail.

Within the mouse embryo, the AVE acts to induce anterior fate, and repress the formation of the primitive streak in the epiblast cells adjacent to it (Rivera-Pérez and Hadjantonakis, 2015). To further determine whether YSLCs possess an AVE-like character, we analysed the ability of YSLCs to inhibit mesoderm and endoderm markers in hESCs in both 2D and 3D. In 3D, hESCs upregulate GATA6 within control conditions (new Figure 6A, B). However, this upregulation is blocked in the presence of YSLC conditioned medium (new Figure 6A, B). As T, a mesoderm marker, was not upregulated in hESCs in control 3D conditions (new Figure 6C, D), we cultured hESCs in a mesoderm differentiation medium (Richter et al., 2014) as a control and also in mesoderm differentiation medium that had been conditioned on YSLCs. This revealed that YSLC conditioned medium was also able to inhibit mesoderm marker upregulation in hESCs (new Figure 6E, F).

Similarly, to further analyse the ability of YSLCs to induce expression of anterior ectoderm markers in hESCs we considered a 2D neural progenitor differentiation protocol. This protocol normally involves dual small molecule targeting of SMAD signalling in hESCs (Chambers et al., 2009; Shi et al., 2012). Given that YSLC conditioned medium was able to inhibit SMAD activity in hESCs, we analysed the ability of YSLC conditioned medium to replace small molecule SMAD inhibitors within this 2D neural differentiation protocol. Accordingly, hESCs were cultured in YSLC conditioned medium, rather than SB431542 and NOGGIN, for 11 days, before being cultured in FGF2 for 4 days (new Figure 7E). After the completion of the 15-day protocol, the anterior ectoderm markers SOX1, PAX6, NESTIN, and NOTCH were all significantly upregulated (new Figure 7F-H).

Overall, these experiments provide further evidence that YSLCs share functional attributes with the mouse AVE in their ability to induce anterior ectoderm markers at the expense of mesoderm and endoderm markers in hESCs. However, we agree that it is not yet clear whether YSLCs resemble a subset of the human yolk sac, or the yolk sac generally. Indeed, the post-implantation yolk sac of the Cynomolgus monkey has been shown to initially express CER1 in almost every cell, before CER1 becomes anteriorly restricted (potentially representing the primate AVE) (Sasaki et al., 2016). Without transcriptional characterisation of the human AVE, it remains unclear whether YSLCs precisely resemble human AVE specifically, or human post-implantation yolk sac generally. Given that the mammalian AVE has been implicated in symmetry breaking along the anterior-posterior axis of the epiblast, future work should focus on how these YSLCs can be used to recapitulate this process using hPSCs in vitro and we have outlined this as a suggestion for future directions in the discussion of the paper.

– Figure 1 panel K: Nanog expression levels are increase after ACL treatment, when Nanog is generally considered a marker of the epiblast. Could the authors explain this discrepancy?

Although ACL treatment of hPSCs leads to the upregulation of endoderm markers, the reviewers are correct that NANOG continues to be co-expressed within treated cells. The inability of ACL to fully downregulate pluripotency markers in hPSCs may indicate that cells are not completely committed the to an endodermal fate, and instead still preserve some features of pluripotent cells. We have now ensured we draw attention to this fact in the result and Discussion sections of the paper and highlight it as an area for future optimisation.

– Figure 2 panel D: Rset-ACL cells have a gene expression signature more similar to hypoblast, while hEPSC-ACL seem to be similar to both extraembryonic and definitive endoderm. Nevertheless, as mentioned before (line 164-165), hEPSCs-ACL are segregated into different subpopulations expressing different levels of Gata6/Sox2. Did the authors consider these differences in the population when performing bulk RNA-seq? Is it possible that eliminating the Gata6low/Sox2high subpopulation improves the induction to extraembryonic endoderm from EPSCs?

As shown in Figure 1 —figure supplement 3, before bulk RNAseq we sorted hESC-ACL, Rset-ACL and hEPSC-ACL populations so that only the SOX17::H2B-tdTomato+ population from each were sequenced.

– Figure 3 panel A-B: In the quantification of Brachyury+ cells during ACL treatment (panel A), in the case of hESCs at day 2 there is about 30% of Brachyury+ cells, however in the immunofluorescence show in panel B for hESCs at day 2 the percentage of Brachyury+ cells seems to be much higher than 30%. Could the authors explain this discrepancy? Maybe a quantification of the relative fluorescence intensity as done in Figure 1 will be helpful to clarify this point, since it might be difficult to discern between positive and negative cells as there is a gradient of signal intensity.

We agree that the image chosen was not representative of the overall result. We observed that T+ cells tend to appear in clusters, meaning some areas have no T+ cells and others have many. This is why in the figure there appears to be more T+ cells than what the quantification shows. Accordingly, we have changed the image so that it is more representative (new Figure 3B). We also provide a quantification of fluorescence intensity as suggested by the reviewer (new Figure 3A).

–Figure 6 panel E and H. The authors showed that culturing hESCs with yolk sac cell-conditioned media in 3D Geltrex settings prevents Sox2 downregulation and promotes a columnar spheroid morphology (line 387 in the text). In figure 6, they tested whether secretion of BMP and Wnt antagonist by yolk sac cells are responsible for preventing pluripotency exit in 3D (line 415 in the text), by treating hESCs with BMP4/Wnt3a and yolk sac-conditioned media in 2D cultures, nevertheless, the author do not comment about what is the effect of these conditions in the 3D context, given that the formation of columnar epithelial spheroids is also promoted with yolk sac-conditioned media. Do the cells form predominantly columnar/squamous spheroids or do they lose their integrity? Also, Sox2 is a known marker for ectodermal differentiation, did the authors test what is the effect of adding BMP4/Wnt3a and yolk sac-conditioned media in other pluripotency markers like Oct4 or Nanog?

We agree with the reviewers that it is important to understand whether BMP and WNT signalling is being perturbed by YSLCs in 3D Geltrex. To answer this question, we have stained for pSMAD1/5 and LEF1 hESCs in 3D Geltrex under control conditions and in the presence of YSLC conditioned medium (new Figure 5F- H). This revealed that both pSMAD1/5 and LEF1 are downregulated in hESCs in 3D Geltrex when cultured in the presence of YSLC conditioned medium relative to control conditions, demonstrating that YSLCs are able to inhibit signalling WNT and BMP signalling pathway activation in hESCs in 3D. We had initially considered adding BMP4/WNT3a to hESCs in 3D Geltrex and determining whether YSLC conditioned medium could block their effects. However, this would not demonstrate whether under control conditions BMP/WNT were the differentiation-inducing factor within Geltrex, only that YSLC conditioned medium could inhibit the BMP/WNT signal that was being exogenously applied. We feel that applying BMP4/WNT3a in hESCs in 2D is a more effective way of determining whether YSLC conditioned medium can prevent BMP/WNT signalling as the differentiation-inducing pathway is previously controlled.

We have now also analysed the effect of BMP4 and WNT3a supplementation in the presence of YSLC conditioned medium on NANOG levels in hESCs (new Figure 5—figure supplement 1H-K). The demonstrated that YSLC conditioned medium could prevent WNT3a-/BMP4- induced downregulation of NANOG, as it does for SOX2.

– Figure 7. The dynamics of ectodermal and pluripotency markers seem insufficient to support the idea of stable induction of anterior ectoderm. Sox1 is high since day 1 and Otx2 goes down at day 5. Is it possible that these markers get stabilized after more than 5 culture days? The authors should analyze the expression of mesodermal and endodermal markers to show that these markers are not induced.

We agree with the reviewers’ comment that claiming YSLCs are able to induce a stable anterior ectoderm fate in hESCs is too strong as a statement. Accordingly, we have altered the language in the paper to clarify that YSLCs are able to induce the upregulation of anterior ectoderm markers in hESCs, rather than a complete anterior ectoderm fate. Without transcriptional data describing human anterior ectoderm, it is not possible to claim that YSLCs can induce it. We also observe that the pluripotency markers OCT4 and NANOG are co-expressed with SOX1 and OTX2 in hESCs cultured in 3D Geltrex in YSLC conditioned medium (new Figure 7A-D and Figure 7—figure supplement 1C). Within mouse development, the AVE acts to antagonise signals from the trophectoderm (Rivera-Pérez and Hadjantonakis, 2015). Therefore, we hypothesize that the addition of trophoblast stem cell may be necessary to downregulate pluripotency markers in the presence of YSLCs. This is an interesting line of future research and we have drawn attention to this in the revised discussion of the paper.

We also analysed the expression of SOX1 after 6 and 7 days in hESCs cultured in 3D Geltrex in the presence of YSLC conditioned medium. This revealed that SOX1 levels are eventually downregulated relative to controls (Figure 7—figure supplement 1A, B). Therefore, YSLCs are not able to induce a full anterior ectoderm fate in hESCs. This again suggests that the addition of trophoblast stem cells may be needed to consolidate an anterior ectoderm fate.

Revisions expected in follow-up work:Additional experimental data in support of the identify of Rec-ACL cells as genuine post implantation yolk sac cells, and discerning their activity as equivalent to the murine AVE.

Overall, we hope that our additional experimental data revealing that YSLCs are able to inhibit the upregulation of mesoderm and endoderm markers, while supporting the expression of ectoderm and pluripotency markers, in hESCs via WNT and BMP signalling pathway inhibition, bolsters our suggestions that YSLCs are yolk-sac like and harbour AVE-like functionality.

References:

Chambers, S. M., Fasano, C. A., Papapetrou, E. P., Tomishima, M., Sadelain, M., and Studer, L. (2009). Highly efficient neural conversion of human ES and iPS cells by dual inhibition of SMAD signaling. *Nature Biotechnology*, *27*(3), 275–280. https://doi.org/10.1038/nbt.1529

Cuomo, A. S. E., Seaton, D. D., McCarthy, D. J., Martinez, I., Bonder, M. J., Garcia-Bernardo, J., … Consortium, H. (2020). Single-cell RNA-sequencing of differentiating iPS cells reveals dynamic genetic effects on gene expression. *Nature Communications*, *11*(1), 810. https://doi.org/10.1038/s41467-020-14457-z

Huang, K., Zhu, Y., Ma, Y., Zhao, B., Fan, N., Li, Y., … Pan, G. (2018). BMI1 enables interspecies chimerism with human pluripotent stem cells. *Nature Communications*, *9*(1), 4649. https://doi.org/10.1038/s41467-018-07098-w

James, D., Noggle, S. A., Swigut, T., and Brivanlou, A. H. (2006). Contribution of human embryonic stem cells to mouse blastocysts. *Developmental Biology*, *295*(1), 90–102. https://doi.org/https://doi.org/10.1016/j.ydbio.2006.03.026

Masaki, H., Kato-Itoh, M., Umino, A., Sato, H., Hamanaka, S., Kobayashi, T., … Nakauchi, H. (2015). Interspecific in vitro assay for the chimera-forming ability of human pluripotent stem cells. *Development*, *142*(18), 3222 LP – 3230. https://doi.org/10.1242/dev.124016

Richter, A., Valdimarsdottir, L., Hrafnkelsdottir, H. E., Runarsson, J. F., Omarsdottir, A. R., Ward-van Oostwaard, D., … Valdimarsdottir, G. (2014). BMP4 promotes EMT and mesodermal commitment in human embryonic stem cells via SLUG and MSX2. *Stem Cells (Dayton, Ohio)*, *32*(3), 636–648. https://doi.org/10.1002/stem.1592

Rivera-Pérez, J. A., and Hadjantonakis, A.-K. (n.d.). The Dynamics of Morphogenesis in the Early Mouse Embryo. *Cold Spring Harbor Perspectives in Biology*, *7*(11), a015867. https://doi.org/10.1101/cshperspect.a015867

Sasaki, K., Nakamura, T., Okamoto, I., Yabuta, Y., Iwatani, C., Tsuchiya, H., … Saitou, M. (2016). The Germ Cell Fate of Cynomolgus Monkeys Is Specified in the Nascent Amnion. *Developmental Cell*, *39*(2), 169–185. https://doi.org/10.1016/j.devcel.2016.09.007

Shi, Y., Kirwan, P., and Livesey, F. J. (2012). Directed differentiation of human pluripotent stem cells to cerebral cortex neurons and neural networks. *Nature Protocols*, *7*(10), 1836–1846. https://doi.org/10.1038/nprot.2012.116

Stirparo, G. G., Boroviak, T., Guo, G., Nichols, J., Smith, A., and Bertone, P. (2018). Integrated analysis of single-cell embryo data yields a unified transcriptome signature for the human pre-implantation epiblast. *Development*, *145*(3), dev158501. https://doi.org/10.1242/dev.158501

Stuart, T., Butler, A., Hoffman, P., Hafemeister, C., Papalexi, E., Mauck, W. M. 3rd, … Satija, R. (2019). Comprehensive Integration of Single-Cell Data. *Cell*, *177*(7), 1888-1902.e21. https://doi.org/10.1016/j.cell.2019.05.031

Xiang, L., Yin, Y., Zheng, Y., Ma, Y., Li, Y., Zhao, Z., … Li, T. (2019). A developmental landscape of 3D-cultured human pre-gastrulation embryos. *Nature*. https://doi.org/10.1038/s41586-019-1875-y

Yang, Y., Liu, B., Xu, J., Wang, J., Wu, J., Shi, C., … Deng, H. (2017). Derivation of Pluripotent Stem Cells with in vivo Embryonic and Extraembryonic Potency. *Cell*, *169*(2), 243-257.e25. https://doi.org/10.1016/j.cell.2017.02.005

Zhou, F., Wang, R., Yuan, P., Ren, Y., Mao, Y., Li, R., … Tang, F. (2019). Reconstituting the transcriptome and DNA methylome landscapes of human implantation. *Nature*, *572*(7771), 660–664. https://doi.org/10.1038/s41586-019-1500-0

[Editors' note: further revisions were suggested prior to acceptance, as described below.]

1. The revised manuscript did not fully convince the reviewers that the ACL protocol represents a novel way to produce human extraembryonic endoderm.Lines 192-194: "This revealed that all three cell populations bore a greater similarity to the hypoblast following ACL^-^treatment, while their similarity to the epiblast decreased, suggesting a shift towards an endodermal fate (Figure 1L)."One is concerned the data simply show that sorting SOX17-positive cells from a mixed population of differentiating stem cells can enrich the endodermal transcriptional signature. Why is it surprising that the SOX17-positive subpopulation of differentiated stem cell lines bears similarity to endoderm, when SOX17 is known to be both necessary and sufficient for endodermal differentiation? In other words, does the novel ACL treatment protocol enhance the endodermal qualities of SOX17-positive cells that can be produced from stem cell lines by any other differentiation protocol? Given that the goal here is to establish a protocol for generating hypoblast-like cells from hESCs and the current methods direct hESCs towards definitive endoderm fate, it would be important to include in Figure 1L also definitive endoderm for comparison.

In response to this reviewer’s comment, we have added in definitive endoderm into New Figure 1L. New Figure 1L uses single cell data from the three human embryo lineages and from an in vitro definitive endoderm derivation protocol. We have replaced the previous Z-score based analysis with the previously published Gene Set Variation Analysis (GSVA) which has been shown to provide robust cell type extrapolations of bulk RNA-sequencing samples based on single-cell RNA sequencing-derived markers (Diaz-Mejia et al., 2019; Hänzelmann, Castelo, and Guinney, 2013). We have also added in a third human hypoblast dataset (Molè et al., 2021) which has been published since our last submission, further strengthening our marker analysis. This demonstrated that hESC-ACL were the most similar to definitive endoderm. Contrastingly, hEPSC-ACL and Rset-ACL were comparatively less similar to this endodermal state. Conversely, the same analysis was carried out comparing the cell lines to the human hypoblast, revealing Rset-ACL to be the most similar to the human hypoblast and hESC-ACL the least similar.

GSVA analysis requires the definition of gene signatures for each lineage used. To do this, we performed pairwise differential gene expression analysis for each lineage of interest compared to each of the others (e.g. Epiblast versus Hypoblast; Epiblast versus Trophoblast, etc.), and selected only those common markers for each of the individual gene signatures, resulting in rigorous marker selection criteria. Since different gene signatures were used for each lineage, the table presented in New Figure 1L can only be read across cell lines, not across lineages, to compare scores. To try and emphasize this, scores were normalized across rows and breaks were placed between them, and this point was emphasized in the text. Therefore, while New Figure 1L clearly shows that ACL treated cell populations show increased similarity to endodermal populations, and that hESC-ACL is more similar to definitive endoderm while Rset-ACL is more similar to hypoblast, it does not answer the question of how cell populations compare across lineages. To answer this question, we performed a direct comparison in New Figure 2 with the updated GSVA protocol (further detail provided in point 3). This revealed that hESC-ACL was more similar to in vitro definitive endoderm. Contrastingly, Rset-ACL was found to be more similar to human extra-embryonic endoderm. Collectively, the analyses in both New Figure 1L and New Figure 2 demonstrate that Rset hPSCs respond to ACL to derive an endodermal lineage that is not similar to the endodermal lineage derived using previously published protocols (definitive endoderm). The converse is true for hESCs, that respond to ACL to derive an endodermal lineage that is very much similar to previously published definitive endoderm derivation protocols.

We found hEPSC-ACL to be overall more similar to in vitro definitive endoderm than to extra-embryonic endoderm, but to an extent that is less extreme than hESC-ACL. This is likely due to the population being heterogenous as is addressed in more detail later in this response.

Please note: owing to her substantial contribution to the resubmitted manuscript in her reanalysing of the RNAseq data, Bailey Weatherbee is now listed as second author and Viviane Souza Rosa is now third. This was agreed upon by the manuscript’s authors.

2. The conclusion that pluripotent state dictates response to the ACL treatment is underdeveloped, owing to flaws in experimental design.In fact, the authors observe major differences between biological replicate hEPSC lines in terms of their response to ACL treatment (compare Figure 1D, top Gata6, Sox17 and Figure 1 Supp 2 K, top Gata6 and Sox17. Additionally, Figure 1 Supp 2 D shows very poor resolution of Sox2 and Gata6 expressing populations after ACL treatment of this cell line). To their credit, the authors note on Line 173: "This suggests that there is a heterogeneous response to ACL^-^treatment within hEPSCs." Yet they then drop this cell line and exclude it from subsequent analyses.Therefore, how can the authors be confident that the reported transcriptional differences captured in Figure 1L (among ACL^-^treated stem cell lines) is meaningful when they haven't really captured variation among biological replicate hEPSC cell lines?In other words, the data shown in Figure 1L are cherry-picked because they exclude the cell line that did not perform as desired (and shown in Supp. 2).

We find this comment surprising because even if there is heterogeneity in the response to ACL treatment between cell lines, they all upregulate endodermal markers at the RNA and protein level and downregulate the pluripotency marker SOX2, in all pluripotent states other than H9 hEPSC (Figure 1 and Old Figure 1—figure supplement 2). We did not sequence H9-ACL treated cells because, as stated, RUES2-GLR contain a SOX17 reporter that allowed us to sort for SOX17+ cells in order to sequence this population. We would have liked to also do the same for H9 where it not for the lack of an endodermal reporter within this cell line.

Nonetheless, to demonstrate that we are not cherry picking our cell lines, and that, indeed, ACL has a universal effect on hPSCs, regardless of the cell line, we have repeated ACL treated on SHEF6-derived hEPSCs, hESCs and Rset hPSCs. As expected, this revealed that endodermal markers were significantly upregulated in all cases both at the RNA and protein level and that SOX2 was significantly downregulated in all cases too (New Figure 1—figure supplement 4).

3. The evidence that the authors have identified a cell type that, on the basis of transcriptome analysis, is a good model for extraembryonic endoderm is also cherry-picked because the authors focus on merely 40 genes to evaluate transcriptional phenotypes. Admittedly, selection of these 40 genes was conducted by rationale that is logical and justified. But the subsequent use of these 40 genes to compare treatments merely ranks the cell lines to each other. The analysis does not produce insight into whether these treatments are wholistically producing extraembryonic endoderm.Moreover, even if we take the authors' word, Line 244 states: "Rset-ACL the most relativelysimilar to extra-embryonic endoderm (Figure. 2C)." Similarly, Line 245 states, "hEPSC-ACL were more similar to extra-embryonic endoderm than to definitive endoderm."Yet examination of Figure 2B does not strongly support these statements. For example, from 20 extra-embryonic endoderm genes only 6 appear to be upregulated, and from 20 definitive endoderm genes 2 are downregulated for Rset-ACL. As the differences in the number of upregulated and down regulated extra embryonic or definitive endoderm genes do not clearly identify a cell type fully aligned with hypoblast, this raises further concerns that the ACL treated cell populations are likely heterogenous rather than representing one YCLC cell type.

Although we included a list of the top 20 markers for both definitive endoderm and extraembryonic endoderm within Old Figure 2, our RNAseq analysis was not confined to just these 40 genes. As outlined within the figure legend and the main text, all differentially expressed genes for both fates were analysed and used within our analysis. This was, therefore, not an example of cherry-picking genes for our analysis.

We agree that the heatmaps provided in Old Figure 2 were misleading and not representative of the RNAseq conclusions and so we have removed them from New Figure 2. Given that the new GVSA RNAseq analysis approach (as outlined above and below) does not use Z-score-based analysis and the highlighting of top 20 markers may mislead readers into thinking these are the only genes analysed, as it confused the reviewers, we have removed both the heatmaps and list of top 20 genes from the figure. Instead, all differentially expressed genes for both human definitive endoderm and human hypoblast can be found in New Supplementary File 2.

For our revised manuscript, we have employed a new method of analysing the relative similarity of ACL-treated cell lines to either hypoblast or definitive endoderm. As outlined above in response to point 1, we used Gene Set Variation Analysis (GSVA), as this methodology can reliably be used to assign cell type labels from single cell RNA-sequencing data to bulk RNA-sequencing data (Hänzelmann et al., 2013; Diaz-Mejia et al., 2019). Once scores were calculated for both gene sets, since the gene sets are inverses of each other (i.e. positive definitive endoderm markers are negative hypoblast markers and vice versa), we subtracted the definitive endoderm score from the hypoblast score and normalized to 1 in order to place the cell populations across a relative scale. Given how similar these two endodermal populations are, placing these populations across a relative scale based on markers is helpful as more wholistic correlative analyses are unlikely to identify the few differences between them, and as well may bias towards the definitive endoderm population provided here given that the definitive endoderm is an in vitro derived cell line, while the hypoblast is embryo-derived. Therefore, we believe this newly provided GSVA analysis provides a robust, published and unbiased approach to address the cell type label transfer between the single-cell data and bulk RNA sequencing data provided here.

We have also added in a third human hypoblast dataset (Molè et al., 2021) which has been published since our last submission, further strengthening our marker analysis. This identified 366 and 1251 markers for the definitive endoderm and the extra-embryonic endoderm, respectively. All of these markers were then used to determine which of the three cell types was the most similar to human extra-embryonic endoderm relative to human definitive endoderm. This revealed that, in harmony with the results from the previous method of analysing, hESC-ACL is the most similar to human definitive endoderm and the least similar to the human hypoblast (New Figure 2C). hEPSC-ACL were found to be overall more similar to the definitive endoderm than the hypoblast, but to a smaller extent that hESC-ACL (New Figure 2C). Once again, Rset-ACL was found to be the most similar to the human extra-embryonic endoderm and the least similar to the definitive endoderm (New Figure 2C). Combined with our pairwise analysis in New Figure 1L. We feel this strongly supports Rset-ACL as the most transcriptionally similar of the three cell types to the human hypoblast.

However, not only do we look at transcriptional similarity, we also look at the dynamics of cell fate specification of these cell types in Figure 3A and B. These results demonstrate that both hEPSC-ACL and hESC-ACL upregulate the mesendoderm marker BRACHYURY, along with SOX17, during ACL treatment, a sequence of expression that is in line with the mesendoderm differentiation that occurs prior to definitive endoderm differentiation of hESCs. Contrastingly, Rset-ACL do not go through this intermediate state, and instead directly convert to an endodermal lineage. This once again demonstrates Rset-ACL to differentiate in a way that is most similar to human hypoblast.

Finally, we also analyse the functional ability of the three cell lineages to contribute to the mouse primitive endoderm. This demonstrated that both Rset-ACL and hEPSC-ACL were able to contribute (Figure 3C-E). However, hESC-ACL were not. Taking into account all of the presented transcriptional, developmental and functional evidence, we can confidently conclude that Rset-ACL is the most similar to the human hypoblast and that hESC-ACL is the most similar to human definitive endoderm, both relatively and in absolute terms. For this reason, only Rset-ACL was taken forward for further analysis within the manuscript.

hEPSC-ACL demonstrate a blend of similarity to both definitive endoderm and hypoblast and likely represent a very heterogenous population. This suggests that the starting hEPSC population may contain subpopulations with largely contrasting pluripotent states that span from naïve to primed. This is mirrored during human trophoblast stem cell induction wherein hEPSCs respond heterogeneously (Castel et al., 2020). It, therefore, does not match the expectation of a cell with full extra-embryonic potential, and given this, we did not consider it a suitable candidate for a human yolk sac-like cell.

4. Chimera experiments are lacking controls.What is the degree chimerism when Rset and hEPSC are not cultured in ACL? Given the very minor degree chimerism (3D and 3G) and the evidence of incomplete population-level differentiation of Rset and hEPSC in ACL (see above), the possibility remains that the authors are simply testing the already-proved hypothesis that Rset and hEPSC can chimerise extraembryonic endoderm in mouse embryos (Gafni et al., 2013; Yang et al., 2017).Moreover, in Figure 3, the number of chimera embryos analyzed nor the number of experiments are not provided.

To answer this query, we carried out interspecies chimeras with Rset hPSCs and hEPSCs. This revealed a starkly different contribution profile to both ACL-treated populations, with Rset hPSCS and hEPSC contributing purely to the epiblast in 81% and 44% of all embryos (New Figure 3—figure supplement 2). Contribution purely to the epiblast did not occur at all for Rset-ACL and hESPC-ACL chimeras. Similarly, there was no example of contribution purely to the primitive endoderm for either hEPSC or Rset hPSCs. This demonstrates that the ability of hEPSC-ACL and Rset-ACL to colonise the primitive endoderm within interspecies chimeras is not an artefact of their previous respective pluripotent state.

The number of embryos and experiments can be found in the respective legends of each figure.

5. Which cell line is the most similar to postimplantation hypoblast? Still unclear.A transcriptional comparison was performed in Figure 4. However, the same weaknesses detailed in Point 3 above also apply here. In short, the transcriptional differences shown do not compellingly support the authors' conclusions.

As described above, in this resubmitted manuscript, a new, more rigorous process was used to draw the same conclusions presented previously. Briefly, to identify markers of either pre- or post-implantation human hypoblast, we still utilized the FindMarkers function in Seurat and used those genes with a minimum Log2 fold change of 0.25, p-value of less than 0.05 (Wilcoxon test), and expressed in at least 10% of cell type of interest to define a gene signature. This identified 397 and 618 markers for the pre-implantation and the post-implantation hypoblast, respectively. All of these markers were then used to determine which of t2ilGo-Pre or Rset-ACL was the most similar to the post-implantation human yolk sac relative to the human pre-implantation extra-embryonic endoderm (primitive endoderm) using the previously described GSVA analysis (see point 1 and point 3).

A functional analysis was performed in Figure 4 Supp. 2 in postimplantation chimeras. However, only a single cell line was evaluated (only the cell line they claim to be the most postimplantation). Therefore, this experiment lacks appropriate controls. Additionally, it is unclear why the authors chose to mimic mouse postimplantation ex vivo, when the standard is to transfer the chimeras to recipient mice to observe actual postimplantation stages.

We agree that the gold standard of this experiment would be to perform embryo transfers. However, we do not have ethical approval to carry out transfers of mouse embryos containing human cells. We, therefore, have removed the post-implantation chimera data from the revised manuscript.

6. Evaluation of "cell fate" in Figure 5 and 6 is superficial.Only a single pluripotency marker (Sox2) and a single mesendoderm marker (T) are analysed. Therefore, the results are still preliminary and conclusions weaker than they should be.In these experiments, gene expression is analyzed primarily at the protein level by relative fluorescence intensity (RFI).

As previously shown, we have analysed a second pluripotency marker, NANOG (Figure 5—figure supplement 1H-K), demonstrating that YSLC conditioned medium is also able to counteract the downregulation of NANOG in hESCs when exposed to both BMP4 and WNT3a in 2D, just as is the case for SOX2. Similarly, the effect of YSLC conditioned medium on NANOG and OCT4 levels in 3D was also explored (Figure 7A and B).

In addition, we have now added qPCR analysis of further posterior epiblast markers, EOMES, SOX17, MIXL1, FOXA2, and GSC (New Figure 6G). This revealed a complementary result to that already observed at the protein level: that YSLC conditioned medium is able to inhibit the upregulation of posterior epiblast markers in hESCs.

As DAPI (or other nuclear marker) images are not shown, is RFI relative to DNA label? This should be clarified.

As outline in the ‘image analysis’ section of our Materials and methods (line 911) we do not normalise to a DNA label.

7. It is not clear how stable over time are YSLC when cultured in ACL?

We have provided new evidence that YSLC are stable over long term culture by demonstrating maintenance of their endodermal fate after 1 month (New Figure 1—figure supplement 3).

The Discussion should more clearly articulate the limitations of the YSLC model and that hEPSCs gave rise to a population of cells that bears a similarity to the extra-embryonic endoderm but may also contain a definitive endoderm- like subpopulation.

We have now added further discussion concerning hEPSC heterogeneity (lines 529-542). We have also now discussed at length the limitations of our model (lines 581-608), covering how human trophoblast cells must be incorporated into our model in 3D and how YSLCs 3D culture conditions must be optimised in order for this to happen. Moreover, we discuss how future studies benchmarking the anterior hypoblast in human embryos will be needed to determine whether the YSLCs capture specifically this subpopulation.